# Adaptive Graph Convolutional Network with Attention Fusion for Multivariate Time Series Forecasting with Variable Missing

## Abstract

Multivariate time series forecasting (MTSF) plays a vital role in diverse applications such as traffic prediction, weather research, and energy management. However, missing subset variable forecasting has emerged as a critical challenge in MTSF due to factors such as sensor failures and maintenance. Variable incompleteness severely hinders the ability of forecasting models to capture intrinsic inter-variable relationships. Existing approaches either suffer from severe error accumulation, lack flexible mechanisms for handling missing data, or overly rely on local spatiotemporal correlations. To address these limitations, we propose VMPredictor, a novel end-to-end framework that effectively models spatiotemporal dependencies among incomplete variables for accurate forecasting. VMPredictor incorporates two key components: (1) the Adaptive Missing Filling and Enhancement Layer , which introduces learnable embeddings to adaptively fill missing positions and dynamically refine incomplete representations during training; and (2) the Spatiotemporal Dependency Mining Layer, built upon a Dynamic Graph Convolution Layer-Normalized Gated Recurrent Unit, where dynamic graph convolution adaptively reconstructs spatial correlations and replaces all fully connected layers in GRU to capture synchronized spatiotemporal dependencies. Together, these innovations endow VMPredictor with robust missing-data handling and precise spatiotemporal relation modeling. Extensive experiments on five real-world datasets under varying missing rates demonstrate the superiority of our approach over existing baselines. Codes can be found at https://anonymous.4open.science/r/Code-A216/.

## 1 Introduction

Multivariate time series forecasting (MTSF) plays a critical role in real-world applications such as traffic flow prediction Yu et al. (2018), air quality assessment Yu et al. (2025a), and energy demand management Khan et al. (2023). Despite substantial progress, MTSF faces several inherent challenges in practical scenarios, especially when dealing with variable missing data. In real-world systems, some sensors or data sources may intermittently fail or be unavailable due to maintenance, extreme weather, or communication disruptions. Consequently, a subset of variables observed during training may be entirely missing at inference, leading to the challenging missing variable-subset forecasting (MVSF) scenario Chauhan et al. (2022). This setting introduces asymmetric knowledge incompleteness, where the model has full information during training but only partial observations during inference. Such asymmetry can disrupt both intra-variable temporal dependencies and inter-variable spatiotemporal interactions, significantly degrading prediction accuracy. Additionally, the severity of the problem increases when critical variables are missing, as naive strategies that discard unavailable variables can severely compromise overall model performance Yu et al. (2024).

A common approach to address MVSF is two-stage forecasting, which first imputes missing values and then performs prediction. Existing imputation techniques leverage intra-variable temporal dependencies and inter-variable correlations to reconstruct missing observations Cini et al. (2021); Kong et al. (2023); Chen et al. (2023b); Zhou et al. (2024; 2025). For instance, DGCRIN Kong et al. (2023) employs fine-grained dynamic graph convolutional networks to model structural temporal

dynamics, while GATGPT Chen et al. (2023b) utilizes a graph attention network to enhance pre-trained models' capacity to capture spatial dependencies for spatiotemporal imputation. However, under MVSF, the structural absence of entire variable subsets during inference fundamentally impairs both temporal and spatial dependency modeling. Moreover, distributional shifts between fully observed training data and partially observed inference scenarios exacerbate imputation errors Liang et al. (2025), and the error accumulation across the two stages can lead to suboptimal predictions.

Beyond two-stage methods, end-to-end approaches directly integrate missingness patterns into forecasting. For example, GCN-M Zuo et al. (2023) uses an attention-based memory network to capture local spatiotemporal and global historical features, while GinAR Yu et al. (2024) combines interpolation attention with adaptive graph convolution to iteratively recover missing variables. More general frameworks, such as TRF Hu et al. (2024), TOI-VSF Hao et al. (2025b), and VIDA Liang et al. (2025), employ generative modeling, self-supervised imputation, or cross-domain knowledge transfer to jointly optimize reconstruction and prediction. Nevertheless, these approaches still face limitations in modeling global spatiotemporal dependencies and maintaining flexibility in highly heterogeneous missing data scenarios.

Motivated by the above observations, in this paper, we propose a Variable Missing Predictor (dubbed VMPredictor) for multivariate time series forecasting, which adaptively learns spatiotemporal dependencies by fully leveraging the historical observations of all available variables. The framework comprises four key components: (1) the Adaptive Missing Filling and Enhancement Layer (AMFE-Layer), which fills missing positions with learnable embeddings to dynamically update incomplete representations during training; (2) the Embedding Layer (EmbLayer), which injects temporal information; (3) the Spatiotemporal Dependency Mining Layer (STDMLayer), built upon the Dynamic Graph Convolution Layer Normalized Gated Recurrent Unit (DGCLNGRU), where dynamic graph convolution adaptively reconstructs spatial correlations among variables and replaces all fully connected layers in GRU to capture synchronized spatiotemporal dependencies; and (4) the Multi-Head Temporal Self-Attention Layer (MHTSALayer), which integrates global spatiotemporal information. In this manner, VMPredictor effectively mitigates the issue of error accumulation. In summary, the main contributions of this work are as follows:

- We present VMPredictor to address the problem of missing variables in prediction tasks. The proposed model features flexible missing variable handling capabilities, effectively addressing the problem of error accumulation during the modeling process.

- We introduce a learnable missing embedding to mitigate the impact of parameter learning bias caused by fixed fill-in values, and carefully design the DGCLNGRU to restore spatial relationships between variables and capture synchronous spatiotemporal dependencies during recursive modeling.

- Experimental validation on five real-world datasets demonstrates that VMPredictor outperforms 10 baseline models across all datasets. Even in the presence of high missingness rates, it still achieves accurate predictions for all variables.

## 2 RELATED WORK

**MTSF methods:** MTSF is a long-studied task with numerous pioneering works contributing to its development. Initially, the most straightforward approach to solving this problem was to employ time series forecasting methods (e.g., recurrent neural networks, RNNs) to accurately model the temporal dependencies within the sequence Tan et al. (2020); Lin et al. (2023). Recently, methods leveraging the powerful global self-attention mechanism of Transformers Vaswani et al. (2017) for forecasting have been proposed Yu et al. (2023); Liu et al. (2023b); Zhang & Yan (2023); Liu et al. (2023a), achieving promising results. To reduce the quadratic computational complexity of the vanilla self-attention mechanism, improved variants such as Zhou et al. (2021); Wu et al. (2021); Zhou et al. (2022) have subsequently been introduced. In addition, STGCN Yu et al. (2018), MT-GNN Wu et al. (2020), DFDGCN Li et al. (2024), and FourierGNN Yi et al. (2023) have focused on improving performance by modeling the spatial dependencies. Currently, lightweight forecasting methods based on multilayer perceptron (MLP) have gained widespread popularity, with models such as STID Shao et al. (2022), DLinear Zeng et al. (2023), TSMixer Ekambaram et al. (2023), and TimeMixer Wang et al. (2024) being proposed in succession. Although these methods have

achieved remarkable progress in MTSF, they are inherently designed for complete datasets and tend to suffer from degraded performance in scenarios with missing data.

**MTSF with missing value methods:** Compared with two-stage approaches, end-to-end methods focus on the non-missing data, aiming to achieve accurate future predictions by enhancing the modeling capability for the observed values. From a temporal perspective, LGnet Tang et al. (2020) based on RNN and TriB-TCN Zhang et al. (2023) based on temporal convolutional network TCN were proposed successively. However, due to neglecting the modeling of spatial relationships, both of them exhibit limited performance. To fully exploit the spatial relationships, GCN-M Zuo et al. (2023), GSTAE Wang et al. (2023), BiTGraph Chen et al. (2023a) and GinAR Yu et al. (2024) use graph neural networks to model spatial structures and cooperate with time information learning tools to model temporal dependencies from a spatiotemporal perspective. But, these methods still have shortcoming in modeling global spatiotemporal dependencies. Currently, general frameworks such as TRF Hu et al. (2024), TOI-VSF Hao et al. (2025b), VIDA Liang et al. (2025), GIMCC Hao et al. (2025a) and Merlin Yu et al. (2025b) are becoming increasingly popular, attempting to build one model to solve the prediction problem under different missing rates in order to improve the practical application capabilities.

## 3 PRELIMINARIES

Given $T$ timestamps and $N$ variates, an observed multivariate time serie can be represented as $\mathcal{X} = \{X_1^{his}, X_2^{his}, \cdots, X_t^{his}, \cdots, X_T^{his}\} \in \mathbb{R}^{T \times N}$. Among them, $X_t^{his} = \{x_t^1, x_t^2, \cdots, x_t^N\} \in \mathbb{R}^{1 \times N}$ denote the data collected by all variates at times point $t$. Due to sensor failure, communication obstruction, or power outage, every variable in $\mathcal{X}$ may contain missing values, we use a masking matrix $\mathcal{M} = \{M_1, M_2, \cdots, M_t, \cdots, M_T\} \in \mathbb{R}^{T \times N}$ with the same dimension as $\mathcal{X}$, as an additional auxiliary missing guidance, to enable the model to distinguish these normal values. The masking matrix is defined as:

$$M_j^i = \begin{cases} 1, & \text{if } x_t^i \text{ is observed.} \\ 0, & \text{if } x_t^i \text{ is missing.} \end{cases} \tag{1}$$

where $x_t^i$ represent the value of $i$-th instance of $X_t^{his}$ at time step $t$. Intuitively, changes in a single variable within a MTS are not only related to its own historical values but may also be associated with other variables. Such dependencies can be represented by a graph $G = \{V, E, A_s\}$, where $N = |V|$ denotes the number of variables, $V$ is the set of dependency relationships, and $A_s \in \mathbb{R}^{N \times N}$ is the matrix representation of the graph. In addition, for the convenience, we use the slice notation $X_{1:T}^{his}$ to denote the values in a time window of size $T$ from time step 1 to $T$.

**Multivariate time series forecasting with variable misising** Given a lookback window $X_{1:T}^{his} \in \mathbb{R}^{T \times N}$ of length $T$ and its corresponding masking matrix $M_{1:T}\mathbb{R}^{T \times N}$, the objecticve of Multivariate time series forecasting with variable missing is to learn a mapping functon $\mathcal{P}$ that generates a future prediction window $Y_{1:F}^{\in}\mathbb{R}^{F \times N}$ of length $F$, denoting as $[X_{1:T}^{his}, M_{1:T}, G] \xrightarrow{\mathcal{P}} Y_{1:F}$. In the training phase, we train the model by reconstructing the observed values. Formally, the loss function of the model is designed as follows:

$$Loss(Y_{1:F}, \hat{Y}_{1:F}) = \frac{\sum_{i=1}^N \sum_{j=1}^T |y_j^i - \hat{y}_j^i|}{\sum_{i=1}^N \sum_{j=1}^T} \tag{2}$$

## 4 METHODOLOGY

The framework of our proposed **VMPredictor** is shown in Figure 1. It comprises the following key modules: the Adaptive Missing Filling Enhancement Layer (AMFELayer), the Embedding Layer (EmbLayer), the Spatiotemporal Dependency Mining Layer (STDMLayer), the Multi-Head Temporal Self-Attention Layer (MHTSALayer) and regression layer. The AMFELayer and EmbLayer are responsible for handling missing values in the input sequence and injecting spatiotemporal embedding information, respectively. The STMLayer aims to restore the spatial relationships between variables and further learn the hidden spatiotemporal dependencies within the data. The MHTSALayer is used to extract the global contextual information of the sequence, and regression layer projects the final prediction results.

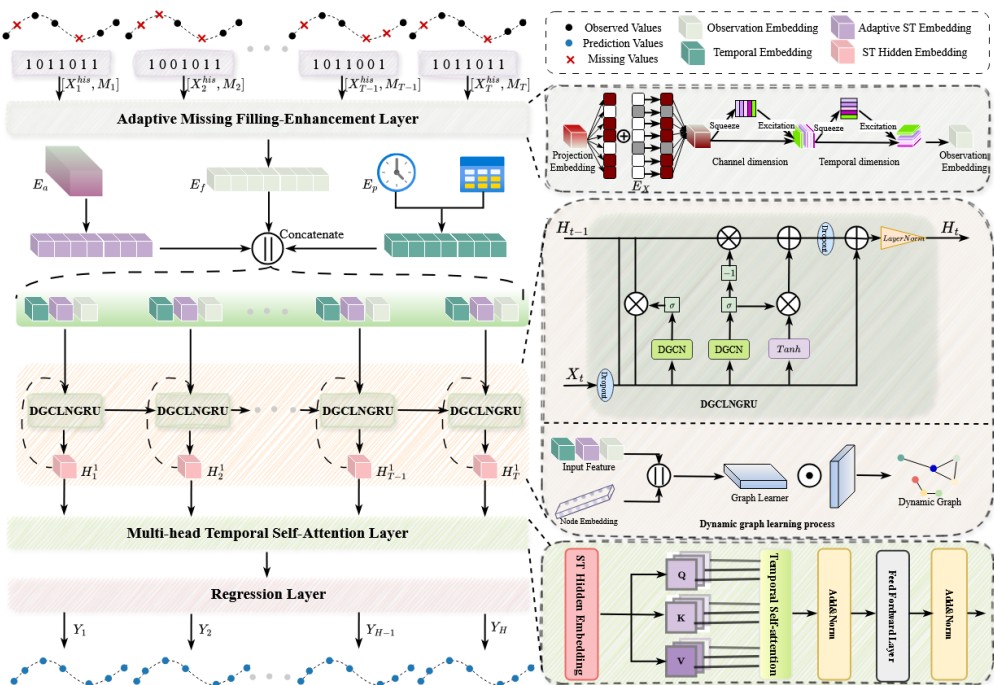

Figure 1: The Framework of our proposed VMPredictor.

## 4.1 ADAPTIVE MISSING FILLING ENHANCEMENT LAYER

The first step in handling incomplete inputs is to effectively address the missing values within the sequence. We decompose this process into two stages: the first stage focuses on adaptive filling of the sequence, while the second stage enhances the filled sequence by augmenting information along both the channel and temporal dimensions.

**Adaptive Filling** In previous studies, missing values in input sequence are typically filled with predefined values (e.g., zeros, last observed values, etc.) or random noise, allowing the model to proces Kong et al. (2023). While intuitive, this approach can reduce model flexibility and may alter the original data distribution, especially when the missing rate is high. For instance, if missing values are filled with zeros and the missing rate exceeds 50%, the resulting data distribution can significantly deviate from the original, which in turn directly affects model prediction performance Zhou et al. (2024). To better handle missing values, we introduce an adaptive missing value filling method based on learnable random embedding parameters $E_X$, which enhances the model's ability to represent incomplete data. Specifically, first, the original sequence $X_{1:T}^{his}$ is projected into a high-dimensional space through a fully connected layer to obtain the initial feature representation $H_{init} \in \mathbb{R}^{T \times N \times d}$. Then based on the masking matrix $M_{1:T}$, $H_{init}$ and $E_X \in \mathbb{R}^{T \times N \times d}$ are fused to generate a comprehensive representation $H_X$ of the input.

$$H_{init} = FC(X_{1:T}) \tag{3}$$

$$H_X = H_{init} \odot M_{1:T} + E_X \odot (1 - M_{1:T}) \tag{4}$$

where $d$ denote the channel dimension, $FC(\bullet)$ indicates a fully connected layer, and $\odot$ represents the Hadamard product.

**Information Enhancement** To fully understand the intrinsic properties of multivariate time series data, feature selection for $H_X$ is essential. As done in Luo et al. (2023), we apply a "squeeze-excitation" (SE) module along both the channel and temporal dimensions to automatically adjust the importance of features in each dimension. In the channel dimension, the SE module consists of two concatenated fully connected layers, and the process for adjusting channel features is as follows:

$$H_{chan} = ReLU(FC(H_X)) \tag{5}$$

$$K_{chan} = Sigmoid(FC(H_{chan})) \tag{6}$$

$$\hat{H}_{chan} = H_X \odot K_{chan} \tag{7}$$

where $ReLU$ and $Sigmoid$ are all non-linear activation functions. Similarly, to focus on the key time point $H_X$, the SE module utilize a double-layer temporal convolution with batch normalization (BN) for feature extraction in the temporal dimension:

$$H_{temp} = Relu(BN(\Theta_{1,k} *_{conv} \hat{H}_{chan})) \tag{8}$$

$$K_{temp} = Sigmoid(BN(\Theta_{1,k} *_{conv} H_{temp})) \tag{9}$$

$$E_f = \hat{H}_{chan} \odot K_{temp} \tag{10}$$

where $\Theta_{1,k}$ and $*_{conv}$ represent the convolution kernel and the 2-dimensional convolution operation respectively.

## 4.2 EMBEDDING LAYER

As previous studies have shown Shao et al. (2022); Liu et al. (2023a), providing additional information embeddings to the model can enhance its learning capability. Therefore, we introduce two different types of spatiotemporal embeddings into the model: temporal information embedding $E_p$ and spatiotemporal adaptive embedding $E_a$. The temporal information embedding $E_p = [E_{day}||E_{week}] \in \mathbb{R}^{T \times N \times 2d}$ includes $E_{day} \in \mathbb{R}^{T \times N \times d}$ and $E_{week} \in \mathbb{R}^{T \times N \times d}$ embeddings, where the former represents the timestamp-of-day embedding and the later represents the day-of-week embedding, and $||$ represents concatenation operation. These two embeddings are designed to capture timestamp information using daily and weekly sampling points, respectively. To generate them, we first construct two learnable embedding dict, $W_{day} \in \mathbb{R}^{288 \times d}$ and $W_{week} \in \mathbb{R}^{7 \times d}$, and then use the original temporal data as indices to obtain the $E_{day} \in \mathbb{R}^{T \times N \times d}$ and $E_{week} \in \mathbb{R}^{T \times N \times d}$. On the other hand, the temporal dynamics across different time series of the same variable, as well as the spatial dynamics between different variables, exhibit complex variations. To model this intricate relationship, we employ a self-learning, trainable parameter embedding $E_a \in \mathbb{R}^{T \times N \times d}$. Finally, the spatiotemporal representation $H_{emb}$ of the input sequence is obtained by concatenating $E_f$, $E_f$ and $E_a$:

$$H_{emb} = E_f||E_p||E_a \in \mathbb{R}^{T \times N \times 4d} \tag{11}$$

## 4.3 SPATIOTEMPORAL DEPENDENCY MINING LAYER

Next, we construct a spatiotemporal dependency mining layer (STDMLayer) to learn the explicit and implicit temporal information and variable relationship in MTS data. The STDMLayer consists of two subcomponents: dynamic graph convolution (DGC), dynamic graph convolution layerNorm GRU (DGCLNGRU).

**DGC** The adjacency graph $A_s$ of multivariate time series defined based on prior knowledge can help the model establish basic spatial relationships. For this, we introduce a dynamic graph $A_d$ based on data itself and variable representation learning in addition to $A_s$. First, we randomly initialize a shared trainable variable embedding $E_{var} \in \mathbb{R}^{N \times d_g}$, then concatenate it with the spatiotemporal representation $H_{emb,t}$ of the MTS data, and finally learn the spatial relationship between variate based on the concatenation result to obtain $A_d$:

$$\Phi = FC([FC(E_{var})||H_{emb,t}]) \tag{12}$$

$$A_d = Softmax(ReLU(Tanh(\Phi\Phi^T))) \tag{13}$$

Based on the predefined graph $A_s$ and dynamic graph $A_d$, we perform convolution operations on them to define DGC:

$$H_k = \alpha H_{emb} + \beta H^{k-1} A_s + \gamma H^{k-1} A_d \tag{14}$$

$$H_{out} = \sum_{i=0}^{K} H^k W^K, H^0 = H_{emb} \tag{15}$$

where $K$ represents the depth of information dissemination, and $\alpha$, $\beta$ and $\gamma$ are hyperparameters which control the relative weighting of information flowing through different computational graph

paths. It is worth noting that on datasets without a predefined graph structure $A_s$, the entire DGC can rely on the dynamic graph $A_d$ for spatiotemporal dependency learning.

**DGCLNGRU** Considering the proven effectiveness of GRU in sequence modeling tasks, we substitute its fully connected operations with DGC to jointly capture synchronous temporal dynamics and spatial dependencies inherent in MTS data Zhao et al. (2019); Kong et al. (2023); Li et al. (2023). Furthermore, to improve robustness and generalization, we incorporate $Dropout$ both before the GRU input and after the gating computations, and apply layer normalization ($LN$) to the final output. Given the data features at time step $H_{emb,t}$, the spatiotemporal hidden state $H_{t,gru}$ at the current can be calculated as follows:

$$H_{t,drop1} = Dropout(H_{emb,t}) \tag{16}$$

$$z_t = Sigmoid(DGC(H_{t,drop1}||H_{t-1,gru}, A_s, A_d)) \tag{17}$$

$$r_t = Sigmoid(DGC(H_{t,drop1}||H_{t-1,gru}, A_s, A_d)) \tag{18}$$

$$c_t = Tanh(DGC(H_{t,drop1}||(r_t \odot H_{t-1,gru})) \tag{19}$$

$$H_t = z_t \odot H_{t-1,gru} + (1 - z_t) \odot c_t \tag{20}$$

$$H_{t,drop2} = Dropout(H_t) \tag{21}$$

$$H_{t,gru} = LN(H_{t,drop1} + H_{t,drop2}) \tag{22}$$

Furthermore, by stacking $L$ layers of DGCLNGRU, the model is able to capture more complex patterns and richer feature representations. Now, when the embedding $H_{emb}$ are fed into the DGCLNGRU in chronological order from $1 \to T$, a spatiotemporal hidden output sequence $H_{gru} = [H_{1,gru}, H_{2,gru}, ..., H_{t,gru}, ..., H_{T,gru}] \in \mathbb{R}^{T \times N \times 4d}$ is gradually generated in a step-by-step manner.

## 4.4 Multi-Head Temporal Self-Attention Layer and Regression Layer

After getting the spatiotemporal hidden sequence $H_{gru}$, a multi-head temporal attention module Vaswani et al. (2017) is used to extract the global context information of $H_{gru}$. Given spatiotemporal hidden representation $H_{gru}$, we first derive the Query $Q$, Key $K$, and Value $V$ matrices through three different linear mapping layers:

$$Q = H_{gru}W_Q, K = H_{gru}W_K, V = H_{gru}W_V \tag{23}$$

where $W_Q$, $W_K$, and $W_V$ are all trainable parameters. Subsequently, $Q$ is multiplied by $K^T$ and normalized to obtain the attention scores across different time steps. Based on this attention matrix, the spatiotemporal hidden representations are then updated through interaction with $V$ as:

$$Atn(Q, K, V) = Softmax(\frac{QK^T}{\sqrt{4d}})V \tag{24}$$

To enhance the perception of temporal semantics across different subspaces, we employ a $L_{tmp}$-layers multi-head temporal attention module to enrich the representation of information, thereby yielding the final spatiotemporal representation $H_{sa}$.

Finally, the output $H_{sa}$ of the MHTSALayer is leveraged to generate then final prediction as follows:

$$Y_{1:F} = FC(H_{sa}) \tag{25}$$

where $Y_{1:F}$ is the prediction.

## 5 Experiment

We evaluate VMPredictor against the state-of-the-art (SOTA) forecasting methods under different variable missing rates on five real-world benchmark datasets. We first assess the predictive performance of different methods using three commonly adopted metrics, namely Mean Absolute Error (MAE), Root Mean Square Error (RMSE), and Mean Absolute Percentage Error (MAPE). We then conduct ablation studies to further validate the effectiveness of the proposed modules.

Table 1: Dataset description.

|  | PEMS08 | PEMS04 | METR-LA | PEMS-BAY | China AQI |
|---|---|---|---|---|---|
| #Variates ($N$) | 170 | 307 | 207 | 325 | 350 |
| #Samples ($T$) | 17856 | 16992 | 34272 | 52116 | 59710 |
| Granularity | 5min | 5min | 5min | 5min | 1h |

## 5.1 EXPERIMENT SETTING

**Datasets** We select five widely used datasets for MTSF, with their statistical information summarized in Table 1. These datasets cover three different categories: two traffic speed datasets (PEMS-BAY, METR-LA), two traffic flow datasets (PEMS04, PEMS08), and one air quality dataset (China AQI).

**Baseline methods** We conduct a comprehensive performance comparison between our proposed VMPredictor and 10 SOTA baselines, which can be grouped into three categories. Specifically, DS-former Yu et al. (2023), and MegaCRN Jiang et al. (2023) are representative forecasting methods designed for spatiotemporal sequence modeling. To account for scenarios with incomplete observations, we also include missing-data-specific methods such as TriD-MAE Zhang et al. (2023), GC-VRNN Xu et al. (2023), BiTGraph Chen et al. (2023a), and GinAR Yu et al. (2024). In addition, two-stage forecasting approaches, including DCRNNLi et al. (2017)+GPT4TSZhou et al. (2023), DFDGCNLi et al. (2024)+TimesNetWu et al. (2022), MTGNNWu et al. (2020)+GRINCini et al. (2021), and FourierGNNYi et al. (2023)+GATGPTChen et al. (2023b), are considered as extra benchmarks to further validate the effectiveness of VMPredictor. The details of baseline methods are presented in Appendix A.

**Implementation details** The experimental setup encompasses several aspects: (1) Model hard parameter configuration: the batch size is 48, the learning rate is 0.001, the history/future window size are both 12, the datasets are uniformly divided into training sets, validation sets and test sets according to the ratio in the reference Yu et al. (2024), and the mask variables are randomly selecting according to missing rates of 25%, 50%, 75%, and 90%; (2) Model soft parameter settings, the number of layers in DGCLNGRU $L_{gru}$ is set to 2, and the number of layers $L_{tmp}$ in MHTSALayer is set to 2. To accommodate the characteristics of different datasets, the values of the channel dimension $d$, the trainable variable embedding $d_s$ and the the propagation depth $K$ of GCN vary across datasets, with detailed settings provided in Appendix B.

## 5.2 OVERALL PERFORMANCE

Table 2 presents the predictive performance of different methods on three datasets (PEMS04, PEMS-BAY, China AQI) at missing rates of 25%, 50%, 75%, and 90%. For brevity, results on the PEMS08 and METR-LA datasets are moved to Appendix C. As shown, our proposed VMPredictor consistently achieves the best performance across the three evaluation metrics in most cases. The performance improvement becomes more pronounced as the variable missing rate increases to 75%-90%, which can be attributed to the designed AMFELayer module and the DGCLNGRU in the spatiotemporal learning module. The former allows the model to flexibly handle missing values in the input, while the latter enhances the model's ability to capture spatiotemporal relationships between variables. Notably, GinAR, a model specifically designed for missing variable value prediction, exhibits significant advantages. However, its limited ability to model cross-time-step interactions restricts its overall effectiveness. BiTGraph achieves the second results across all baseline methods at various missing rates. In contrast, our proposed VMPredictor consistently outperforms all methods, maintaining superior results across all scenarios.

## 5.3 ABLATION STUDY

In this section, we conduct ablation studies to evaluate the impact of different components—namely, the adaptive filling, the information Enhancement (IE), temporal embeddings $E_p$, static graph $A_s$, dynamic graph $A_d$, temporal attention layer, and the all embeddings (EM)—on the overall model performance. The results are reported in Table 3. Several conclusions can be drawn: (1) Compared

Table 2: The forecasting performance of different methods.

| Method (r = 25%) | PEMS04 | | | PEMS-BAY | | | China AQI | | |
|---|---|---|---|---|---|---|---|---|---|
| | MAE | RMSE | MAPE | MAE | RMSE | MAPE | MAE | RMSE | MAPE |
| DSformer | 32.86 | 50.15 | 20.88 | 2.91 | 6.32 | 7.15 | 17.73 | 28.22 | 44.81 |
| MegaCRN | 28.26 | 42.22 | 20.18 | 2.85 | 5.93 | 7.03 | 15.32 | 28.41 | 33.35 |
| DCRNN$_G$ | 25.17 | 40.03 | 17.77 | 2.63 | 5.54 | 6.24 | 15.14 | 28.48 | 32.18 |
| MTGNN$_R$ | 24.84 | 39.67 | 18.71 | 2.65 | 5.78 | 5.73 | 14.89 | 27.13 | 33.29 |
| FourierGNN$_A$ | 25.58 | 40.92 | 19.61 | 2.40 | 5.21 | 5.46 | 14.65 | 27.17 | 32.82 |
| DFDGCN$_T$ | 24.43 | 39.48 | 17.40 | 2.58 | 5.39 | 6.17 | 14.62 | 26.33 | 29.73 |
| GC-VRNN | 23.57 | 39.75 | 16.82 | 2.39 | 4.93 | 5.37 | 14.66 | 26.88 | 31.88 |
| TriD-MAE | 24.15 | 39.83 | 16.98 | 2.46 | 5.17 | 5.48 | 14.51 | 26.18 | 29.09 |
| BiTGraph | 23.01 | 38.94 | 16.75 | 2.17 | 4.52 | 5.16 | 13.85 | 25.79 | 28.94 |
| GinAR | 22.52 | 38.22 | 16.45 | 2.10 | 4.34 | 4.90 | 13.72 | 25.51 | 28.27 |
| VMPredictor | **19.75** | **33.34** | **13.72** | **1.77** | **3.99** | **4.00** | **12.49** | **25.14** | **23.96** |

| Method (r = 50%) | PEMS04 | | | PEMS-BAY | | | China AQI | | |
|---|---|---|---|---|---|---|---|---|---|
| | MAE | RMSE | MAPE | MAE | RMSE | MAPE | MAE | RMSE | MAPE |
| DSformer | 33.31 | 51.51 | 21.35 | 3.08 | 6.46 | 7.73 | 19.06 | 30.35 | 48.09 |
| MegaCRN | 31.48 | 47.07 | 21.29 | 3.02 | 7.36 | 7.75 | 16.51 | 29.52 | 35.86 |
| DCRNN$_G$ | 26.56 | 41.64 | 18.21 | 2.87 | 6.22 | 6.75 | 16.82 | 30.83 | 35.24 |
| MTGNN$_R$ | 26.95 | 41.91 | 19.18 | 2.83 | 6.19 | 6.55 | 16.37 | 30.02 | 37.82 |
| FourierGNN$_A$ | 27.31 | 42.35 | 20.55 | 2.71 | 5.93 | 5.96 | 16.13 | 30.77 | 38.37 |
| DFDGCN$_T$ | 26.09 | 41.18 | 18.49 | 2.74 | 6.17 | 7.03 | 15.76 | 29.30 | 32.85 |
| GC-VRNN | 26.43 | 41.34 | 17.83 | 2.64 | 5.34 | 5.86 | 15.70 | 28.67 | 34.21 |
| TriD-MAE | 25.89 | 40.65 | 17.52 | 2.69 | 5.53 | 6.12 | 15.74 | 28.96 | 32.94 |
| BiTGraph | 24.15 | 40.03 | 17.34 | 2.44 | 5.06 | 6.07 | 14.52 | 27.45 | 31.36 |
| GinAR | 23.78 | 39.02 | 17.04 | 2.35 | 4.78 | 5.88 | 14.33 | **26.81** | 29.56 |
| VMPredictor | **20.66** | **34.19** | **14.47** | **1.92** | **4.24** | **4.35** | **13.56** | 26.97 | **25.96** |

| Method (r = 75%) | PEMS04 | | | PEMS-BAY | | | China AQI | | |
|---|---|---|---|---|---|---|---|---|---|
| | MAE | RMSE | MAPE | MAE | RMSE | MAPE | MAE | RMSE | MAPE |
| DSformer | 37.28 | 54.91 | 23.25 | 3.45 | 8.15 | 9.06 | 20.63 | 33.22 | 52.05 |
| MegaCRN | 33.58 | 47.95 | 22.03 | 3.35 | 7.75 | 8.77 | 19.66 | 33.09 | 48.96 |
| DCRNN$_G$ | 28.54 | 43.71 | 19.18 | 3.09 | 7.14 | 7.82 | 17.82 | 31.99 | 37.22 |
| MTGNN$_R$ | 28.04 | 44.36 | 20.90 | 3.02 | 7.06 | 7.75 | 17.94 | 31.97 | 39.29 |
| FourierGNN$_A$ | 29.87 | 45.17 | 22.20 | 2.98 | 6.85 | 7.58 | 18.01 | 31.86 | 39.90 |
| DFDGCN$_T$ | 28.29 | 42.81 | 19.91 | 3.07 | 6.96 | 7.59 | 17.85 | 30.51 | 38.68 |
| GC-VRNN | 27.72 | 43.82 | 18.67 | 2.87 | 6.08 | 6.94 | 16.99 | 30.57 | 37.66 |
| TriD-MAE | 26.95 | 41.90 | 18.04 | 2.79 | 5.97 | 6.85 | 16.79 | 29.84 | 35.76 |
| BiTGraph | 26.33 | 41.69 | 17.92 | 2.61 | 5.79 | 6.68 | 15.62 | 29.01 | 33.58 |
| GinAR | 25.98 | 41.53 | 17.58 | 2.54 | 5.48 | 6.17 | 15.39 | **27.96** | 31.86 |
| VMPredictor | **21.62** | **35.90** | **15.12** | **2.11** | **4.67** | **4.93** | **14.95** | 28.32 | **30.11** |

| Method (r = 90%) | PEMS04 | | | PEMS-BAY | | | China AQI | | |
|---|---|---|---|---|---|---|---|---|---|
| | MAE | RMSE | MAPE | MAE | RMSE | MAPE | MAE | RMSE | MAPE |
| DSformer | 40.31 | 59.18 | 25.62 | 3.72 | 9.06 | 10.23 | 23.17 | 35.72 | 57.31 |
| MegaCRN | 36.14 | 52.17 | 23.42 | 3.54 | 8.25 | 9.23 | 22.61 | 35.74 | 53.28 |
| DCRNN$_G$ | 31.42 | 46.17 | 21.27 | 3.31 | 7.69 | 8.82 | 21.78 | 34.28 | 50.64 |
| MTGNN$_R$ | 30.61 | 45.88 | 21.04 | 3.22 | 7.26 | 8.45 | 20.02 | 34.15 | 50.23 |
| FourierGNN$_A$ | 32.16 | 48.83 | 23.65 | 3.28 | 7.44 | 9.07 | 20.18 | 34.17 | 50.94 |
| DFDGCN$_T$ | 30.98 | 45.93 | 21.43 | 3.27 | 7.15 | 8.34 | 21.06 | 33.19 | 51.28 |
| GC-VRNN | 30.06 | 45.12 | 20.43 | 3.12 | 7.32 | 7.94 | 19.24 | 32.91 | 48.73 |
| TriD-MAE | 29.54 | 44.23 | 20.05 | 3.02 | 7.02 | 7.63 | 18.04 | 32.68 | 45.76 |
| BiTGraph | 28.71 | 43.37 | 19.08 | 2.83 | 6.75 | 7.42 | 17.06 | 31.85 | 38.17 |
| GinAR | 28.20 | 42.82 | 18.31 | 2.77 | 6.43 | 6.94 | 16.83 | **30.97** | 36.30 |
| VMPredictor | **22.18** | **36.81** | **15.61** | **2.28** | **4.96** | **5.20** | **16.40** | 31.77 | **32.06** |

with the zero-prefill method, the adaptive filling method has higher prediction accuracy, which shows that the latter can enable the model to produce relatively reasonable missing data representations for the prediction task. (2) Temporal embeddings play a crucial role in prediction accuracy, and their importance becomes more pronounced as the missing rate increases. This can be attributed to the additional temporal information providing effective guidance for the model. (3) The temporal attention layer also makes a notable contribution, as it enables the fusion of hidden representations across different time steps, thereby offering richer information to the regression layer. (4) The augmentation layer leads to a moderate performance improvement. (5) Removing either the static or dynamic graph results in decreased prediction accuracy. While the static graph captures local spatial dependencies among variables, the dynamic graph further restores and enriches the spatial relationships between them. (6) When we remove all embedding variables $E_p$, $E_X$ and $E_a$, we find that at low missing rates, removing all embedding variables causes a faster decline in model performance than removing $E_p$ while at high missing rates, the opposite is true. This is because when the observed data is sufficient, the model relies more on structural representations $E_X$ and $E_a$, while when observation data is insufficient, structural representations implicitly introduce noise. This indirectly reflects the importance of various embeddings. In addition to the ablation study, we present parameter sensitivity analyses in Appendix E, including the effects of channel dimension $d$, propagation depth $K$ in DGC, and the number of temporal attention layers $L_{temp}$. Appendix D provides visualizations of prediction curves under different missing rates across datasets, and Appendix F reports the model complexity analysis.

Table 3: The result of ablation studies on PEMS04 and PEMS-BAY under the missing rates of 25% and 90%.

| Missing Rate | Model | PEMS04 | | | PEMS-BAY | | |
|---|---|---|---|---|---|---|---|
| | | MAE | RMSE | MAPE(%) | MAE | RMSE | MAPE(%) |
| 25% | zero prefill | 20.49 | 33.98 | 14.32 | 1.78 | **3.96** | 4.01 |
| | w/o. IE | 19.86 | 33.48 | 13.55 | 1.81 | 4.09 | 4.16 |
| | w/o. $E_p$ | 21.34 | 34.47 | 14.75 | 1.83 | 4.11 | 4.11 |
| | w/o. $A_s$ | 19.92 | 33.37 | 14.05 | 1.79 | 3.99 | 4.04 |
| | w/o. $A_d$ | 20.00 | 33.35 | **13.44** | 1.81 | 4.07 | 4.11 |
| | w/o. MHTSA | 20.96 | 34.44 | 15.39 | 1.80 | 3.99 | 4.06 |
| | w/o. EM | 21.23 | 34.53 | 14.83 | 1.84 | 4.14 | 4.35 |
| | VMPredictor | **19.75** | **33.34** | 13.72 | **1.77** | 3.99 | **4.00** |
| 90% | zero prefill | **21.90** | 36.84 | 15.89 | 2.34 | 5.14 | 5.34 |
| | w/o. IE | 21.95 | **36.40** | 16.03 | 2.41 | 5.11 | 5.52 |
| | w/o. $E_p$ | 25.82 | 40.10 | 18.71 | 2.47 | 5.29 | 5.77 |
| | w/o. $A_s$ | 23.19 | 37.43 | 17.14 | 2.30 | 5.00 | 5.23 |
| | w/o. $A_d$ | 23.51 | 38.15 | 18.59 | 2.42 | 5.15 | 5.50 |
| | w/o. MHTSA | 23.05 | 37.43 | 17.14 | 2.46 | 5.17 | 5.62 |
| | w/o. EM | 23.63 | 38.07 | 16.93 | 2.46 | 5.16 | 5.63 |
| | VMPredictor | 22.18 | 36.81 | **15.61** | **2.28** | **4.96** | **5.20** |

## 6 CONCLUSION

In this paper, we propose VMPredictor for multivariate time series forecasting under variable-missing scenarios. VMPredictor incorporates an adaptive missing-value handling mechanism to mitigate the learning bias caused by fixed-value imputation. Furthermore, we design a customized adaptive spatiotemporal dependency learning module, DGCLNGRU, to jointly capture temporal dynamics and spatial structures. Extensive experiments on five real-world benchmark datasets demonstrate the superior performance of VMPredictor under various missing rates. Ablation studies further validate the effectiveness of the proposed adaptive modules. In the future, we plan to explore employing the Transformer architecture as the backbone for spatial modeling, aiming to better capture global spatial dependencies and enhance the practical utility of the model.

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

## A EXPERIMENTAL DETAILS

### A.1 DETAILS OF BASELINE MODELS

The details of the baseline models are briefly summarized as follows. Some experimental results are directly from reference Yu et al. (2024).

- **DSformer**: A prediction method that leverages a dual-sampling module and a temporal variable attention module to capture both global-local characteristics and spatiotemporal dependencies within sequential data.
- **MegaCRN**: A spatiotemporal forecasting method that integrates a Meta-Graph Learner, driven by a Meta Node Bank, into graph convolutional recurrent networks.
- **DCRNN+GPT4TS**: A two-stage forecasting method that employs GPT4TS for data imputation, followed by DCRNN for prediction based on the imputed results.
- **MTGNN+GRIN**: A two-stage forecasting method that employs GRIN for data imputation, followed by MTGNN for prediction based on the imputed results.
- **FourierGNN+GATGPT**: A two-stage forecasting method that employs GATGPT for data imputation, followed by FourierGNN for prediction based on the imputed results.
- **DFDGCN+TimesNet**: A two-stage forecasting method that employs TimesNet for data imputation, followed by DFDGCN for prediction based on the imputed results.
- **GC-VRNN**: It integrates the Multi-Space GNN with the Conditional Variational RNN to perform time series forecasting under missing values.
- **TriD-MAE**: A general pre-trained model for MTSF with missing values based on TCN, which employs a collaborative dynamic position embedding mechanism to improve the model's ability to handle missing patterns.
- **BiTGraph**: It utilizes the Multi-Scale Instance PartialTCN and Biased GCN to jointly captures the temporal dependencies and spatial structure.
- **GinAR**: A carefully designed forecasting model that employs interpolation attention and adaptive graph convolution to recover spatiotemporal dependencies among all missing variables for accurate multivariate time series modeling.

## B PARAMETER SETTINGS

The details of parameter setting of channel dimension $d$, the trainable variable embedding $d_s$ and the propagation depth of GCN $K$, on different datasets are presented in Table 4. We summarize our design principles and provide unified guidelines for adapting the proposed method to different datasets. In our framework, the dimensions $d$ and $d_s$ control the capacity of temporal feature extraction and structural embedding. These values are adjusted according to dataset scale and complexity:for datasets with larger spatial coverage or richer temporal dynamics (e.g., PEMS04, China AQI), moderately larger $d$ and $d_s$ enable the model to capture more expressive patterns, whereas for smaller or less complex datasets (e.g., METR-LA, PEMS-BAY), we reduce these dimensions to avoid over-parameterization and ensure training stability. The spatial hop parameter $K$ determines the range of spatial information aggregation, and its optimal value is similarly dataset-dependent: datasets with dense sensor layouts or strong long-range correlations benefit from a slightly larger $K$, while those with sparse or weak spatial dependencies achieve better performance with smaller $K$. In practice, we find these hyperparameters to be relatively insensitive, and a coarse search within a small range is typically sufficient to achieve robust performance.

Table 4: Hyperparameter settings for different datasets

|  | PEMS04 | PEMS08 | METR-LA | PEMS-BAY | China AQI |
|---|---|---|---|---|---|
| $d$ | 24 | 24 | 16 | 16 | 16 |
| $d_s$ | 20 | 20 | 16 | 16 | 20 |
| $K$ | 3 | 2 | 1 | 1 | 2 |

## C    More Comparison

Table 5 demonstrates the results on PEMS08 and METR-LA dataset under missing rate 25%, 50%, 75% and 90%. The table clearly demonstrates that our proposed VMPredictor has significant advantages, particularly on PEMS08. Among the comparison models, BiTGraph and GinAR outperform the other baseline models, achieving comparable performance. The performance of the pure prediction method is inferior to the two-stage approach, suggesting that imputing missing values can be beneficial to the prediction task.

In addition, to further verify the generalization ability of VMPredictor, we conducted comparative prediction experiments on the PEMS04 and PEMS08 datasets under scenarios of Random Missing (RM) and Block Missing (BM). The experimental results are shown in the Table 6, Table 7. As observed, VMPredictor consistently achieves the best performance, further demonstrating its superior effectiveness in handling missing-data prediction tasks.

## D    Prediction Visualization

In this section, we present prediction heatmaps of our method on three datasets (PEMS04, METR-LA, and China AQI) under $r = 25\%\&90\%$, , providing spatiotemporal evidence of VMPredictor's effectiveness. As shown in the Figure 2, our proposed method generates results that closely match the true data distribution across all four datasets, both under low missing rate $r = 25\%$ and and high missing rate $r = 90\%$, demonstrating the effectiveness of VMPredictor.

## E    Hyperparameter sensitivity

In this section, we evaluate the hyperparameter sensitivity of our proposed model with respect to the channel dimension $d$, the propagation depth $K$ in DGC, as well as the number layer $L_{tmp}$ of MHSALayer on the PEMS08 and METR-LA dataset under the missing rate 50% and 75%.

Figure 3 and Figure 4 show the performance. We first evaluated the channel dimension $d$ on two datasets. As shown in Fig. 3(a, b) and Fig. 4(a, b), the performance metrics exhibit a "decrease-then-increase" trend with respect to $d$, This indicates that when $d$ is too small, the model suffers from insufficient representation and modeling capacity, whereas an excessively large $d$ introduces redundant information and noise, thereby impairing prediction performance. We then evaluate the propagation depth $K$. It can be observed that VMPredictor achieves its best performance with $K = 2$ on PEMS08 and $K = 1$ on METR-LA. This indicates that a deeper information propagation depth does not necessarily lead to better results; an excessively large $K$ may cause the final node representations to become overly similar, thereby hindering effective information discrimination. We then further assess the impact of the $L_{tmp}$. The variation of $L_{tmp}$ on PEMS08 remains stable, with the model achieving its best performance at $L_{tmp} = 2$. In contrast, on METR-LA, the optimal value of $L_{tmp}$ differs when $r = 50\%$. while at $r = 75\%$, the performance tends to stabilize at $L_{tmp} = 2$. This behavior may be attributed to oscillations caused by global temporal information.

## F    Model Complexity Analysis

In this section, we evaluate the model complexity of BiTGraph, GinAR, and VMPredictor across three datasets (PEMS04, PEMS08 and METR-LA) from the perspectives of memory usage, parameter size, and training speed. The experiments are conducted on an Intel(R) Xeon(R) Platinum 8352V CPU @ 2.10GHz with an RTX 4090 GPU, while ensuring a consistent batch size across all models. The results are summarized in Table 8. As shown in the table, the memory usage and training speed of VMPredictor lie between those of BiTGraph and GinAR. The increase in parameter size is mainly attributed to the introduction of the multi-head self-attention mechanism. BiTGraph exhibits the lowest values across all metrics, which can be explained by its convolution-based architecture. In contrast, GinAR incurs higher memory usage and slower training speed due to the graph attention computation. Overall, VMPredictor achieves the best predictive performance with an acceptable level of model complexity.

Table 5: The forecasting performance of different methods on PEMS08 and METR-LA.

| Method | PEMS08 | | | METR-LA | | |
|---|---|---|---|---|---|---|
| ($r = 25\%$) | MAE | RMSE | MAPE | MAE | RMSE | MAPE |
| DSformer | 27.74 | 38.38 | 19.24 | 4.02 | 7.69 | 11.05 |
| MegaCRN | 26.04 | 39.29 | 17.42 | 3.81 | 7.43 | 10.47 |
| DCRNN$_G$ | 24.64 | 36.58 | 15.96 | 3.78 | 7.41 | 10.72 |
| MTGNN$_R$ | 24.08 | 36.65 | 15.17 | 3.69 | 7.28 | 10.48 |
| FourierGNN$_A$ | 23.77 | 35.54 | 15.35 | 3.71 | 7.40 | 10.87 |
| DFDGCN$_T$ | 24.26 | 36.58 | 15.96 | 3.72 | 7.42 | 10.42 |
| GC-VRNN | 23.25 | 35.17 | 14.69 | 3.68 | 7.04 | 10.51 |
| TriD-MAE | 21.53 | 33.15 | 14.25 | 3.64 | 7.15 | 10.37 |
| BiTGraph | 20.65 | 31.89 | 14.05 | 3.61 | 6.74 | 10.25 |
| GinAR | 20.41 | 31.34 | 13.76 | 3.56 | 6.55 | 10.12 |
| VMPredictor | **16.19** | **28.54** | **11.03** | **3.09** | **6.33** | **8.53** |

| Method | PEMS08 | | | METR-LA | | |
|---|---|---|---|---|---|---|
| ($r = 50\%$) | MAE | RMSE | MAPE | MAE | RMSE | MAPE |
| DSformer | 30.47 | 42.49 | 23.79 | 4.38 | 8.27 | 12.78 |
| MegaCRN | 30.68 | 43.43 | 21.30 | 3.94 | 7.87 | 11.02 |
| DCRNN$_G$ | 27.96 | 41.79 | 18.73 | 3.98 | 7.91 | 11.61 |
| MTGNN$_R$ | 25.78 | 37.64 | 17.41 | 3.77 | 7.43 | 11.22 |
| FourierGNN$_A$ | 25.53 | 37.42 | 16.96 | 3.82 | 7.84 | 11.75 |
| DFDGCN$_T$ | 25.39 | 39.05 | 19.42 | 3.89 | 7.68 | 11.45 |
| GC-VRNN | 24.27 | 36.40 | 15.85 | 3.87 | 7.73 | 10.98 |
| TriD-MAE | 23.18 | 35.95 | 15.32 | 3.79 | 7.58 | 11.07 |
| BiTGraph | 22.44 | 35.06 | 14.62 | 3.69 | 7.32 | 10.79 |
| GinAR | 22.01 | 34.53 | 14.21 | 3.61 | 7.14 | 10.42 |
| VMPredictor | **17.01** | **30.16** | **10.96** | **3.22** | **6.69** | **9.10** |

| Method | PEMS08 | | | METR-LA | | |
|---|---|---|---|---|---|---|
| ($r = 75\%$) | MAE | RMSE | MAPE | MAE | RMSE | MAPE |
| DSformer | 35.21 | 51.57 | 25.18 | 4.77 | 9.21 | 14.59 |
| MegaCRN | 32.23 | 48.37 | 21.34 | 4.24 | 8.28 | 12.13 |
| DCRNN$_G$ | 29.62 | 44.82 | 19.79 | 4.15 | 8.16 | 11.93 |
| MTGNN$_R$ | 27.45 | 40.51 | 18.52 | 4.12 | 8.05 | 11.58 |
| FourierGNN$_A$ | 26.89 | 39.44 | 17.95 | 4.21 | 8.25 | 12.25 |
| DFDGCN$_T$ | 28.30 | 42.67 | 20.74 | 4.14 | 8.11 | 11.75 |
| GC-VRNN | 26.32 | 39.67 | 16.06 | 4.17 | 8.19 | 11.71 |
| TriD-MAE | 24.89 | 37.64 | 15.58 | 3.92 | 7.92 | 11.13 |
| BiTGraph | 23.38 | 36.98 | 15.04 | 3.74 | 7.63 | 11.04 |
| GinAR | 23.10 | 36.04 | 14.77 | 3.70 | 7.39 | 10.71 |
| VMPredictor | **18.98** | **34.25** | **11.97** | **3.27** | **6.79** | **9.30** |

| Method | PEMS08 | | | METR-LA | | |
|---|---|---|---|---|---|---|
| ($r = 90\%$) | MAE | RMSE | MAPE | MAE | RMSE | MAPE |
| DSformer | 38.79 | 55.34 | 32.61 | 5.02 | 10.15 | 17.69 |
| MegaCRN | 34.52 | 52.75 | 24.64 | 4.58 | 8.72 | 13.54 |
| DCRNN$_G$ | 31.78 | 46.22 | 22.96 | 4.29 | 8.31 | 12.18 |
| MTGNN$_R$ | 29.15 | 42.79 | 21.33 | 4.25 | 8.29 | 12.14 |
| FourierGNN$_A$ | 29.11 | 41.44 | 20.61 | 4.33 | 8.37 | 12.28 |
| DFDGCN$_T$ | 30.46 | 45.83 | 21.59 | 4.34 | 8.33 | 12.24 |
| GC-VRNN | 28.46 | 41.98 | 20.54 | 4.32 | 8.35 | 12.29 |
| TriD-MAE | 26.18 | 39.25 | 16.43 | 4.11 | 8.22 | 11.29 |
| BiTGraph | 25.01 | 39.06 | 16.18 | 3.91 | 8.03 | 11.78 |
| GinAR | 24.83 | **38.87** | 15.82 | 3.87 | 7.84 | 11.25 |
| VMPredictor | **21.72** | 39.64 | **13.90** | **3.61** | **7.66** | **10.81** |

Table 6: The forecasting performance of different methods on PEMS04 under RM and BM.

| Method | | PEMS04 (RM) | | | PEMS04 (BM) | |
| (r = 25%) | MAE | RMSE | MAPE | MAE | RMSE | MAPE |
|---|---|---|---|---|---|---|
| BiTGraph | 19.47 | 31.61 | **13.39** | 21.41 | **33.98** | 14.97 |
| GinAR | 20.24 | 33.21 | 15.03 | 23.61 | 36.42 | 16.27 |
| VMPredictor | **19.43** | **30.98** | 13.64 | **20.35** | 34.38 | **13.60** |

| Method | | PEMS04 (RM) | | | PEMS04 (BM) | |
| (r = 50%) | MAE | RMSE | MAPE | MAE | RMSE | MAPE |
|---|---|---|---|---|---|---|
| BiTGraph | 20.48 | 32.49 | 14.55 | 24.02 | 37.49 | 16.62 |
| GinAR | 21.37 | 35.53 | 16.22 | 25.77 | 38.22 | 18.04 |
| VMPredictor | **19.77** | **31.75** | **13.79** | **23.05** | **36.50** | **15.56** |

| Method | | PEMS04 (RM) | | | PEMS04 (BM) | |
| (r = 75%) | MAE | RMSE | MAPE | MAE | RMSE | MAPE |
|---|---|---|---|---|---|---|
| BiTGraph | 21.53 | 34.49 | **14.58** | 25.95 | 40.09 | 17.66 |
| GinAR | 24.72 | 38.42 | 18.77 | 26.88 | 41.73 | 19.41 |
| VMPredictor | **21.49** | **33.69** | 15.08 | **24.26** | **38.17** | **17.05** |

| Method | | PEMS04 (RM) | | | PEMS04 (BM) | |
| (r = 90%) | MAE | RMSE | MAPE | MAE | RMSE | MAPE |
|---|---|---|---|---|---|---|
| BiTGraph | 24.13 | 37.71 | 16.28 | 27.29 | 41.91 | 18.38 |
| GinAR | 26.08 | 40.01 | 20.11 | 28.78 | 43.51 | 20.43 |
| VMPredictor | **22.90** | **36.54** | **15.82** | **25.37** | **39.07** | **17.63** |

Table 7: The forecasting performance of different methods on PEMS08 under RM and BM.

| Method | PEMS08 (RM) | | | PEMS08 (BM) | | |
|---|---|---|---|---|---|---|
| $(r = 25\%)$ | MAE | RMSE | MAPE | MAE | RMSE | MAPE |
| BiTGraph | 16.29 | 25.54 | 10.22 | 18.81 | 30.25 | 11.94 |
| GinAR | 20.28 | 32.24 | 14.61 | 21.51 | 33.08 | 21.51 |
| VMPredictor | **15.25** | **24.52** | **9.91** | **17.35** | **28.39** | **11.33** |
| Method | PEMS08 (RM) | | | PEMS08 (BM) | | |
| $(r = 50\%)$ | MAE | RMSE | MAPE | MAE | RMSE | MAPE |
| BiTGraph | 16.88 | 26.30 | **10.82** | 22.74 | 37.07 | 13.83 |
| GinAR | 21.74 | 35.07 | 16.76 | 25.96 | 40.07 | 19.83 |
| VMPredictor | **16.14** | **25.31** | 10.91 | **20.35** | **34.38** | **12.60** |
| Method | PEMS08 (RM) | | | PEMS08 (BM) | | |
| $(r = 75\%)$ | MAE | RMSE | MAPE | MAE | RMSE | MAPE |
| BiTGraph | 18.40 | 28.82 | 12.13 | 26.83 | 42.84 | 15.99 |
| GinAR | 24.39 | 39.53 | 18.91 | 29.57 | 46.18 | 19.92 |
| VMPredictor | **17.24** | **27.52** | **11.60** | **23.37** | **39.07** | **14.63** |
| Method | PEMS08 (RM) | | | PEMS08 (BM) | | |
| $(r = 90\%)$ | MAE | RMSE | MAPE | MAE | RMSE | MAPE |
| BiTGraph | 21.55 | 33.94 | 14.36 | 30.08 | 46.56 | 18.08 |
| GinAR | 26.21 | 42.38 | 20.14 | 29.69 | 46.03 | 21.39 |
| VMPredictor | **19.72** | **32.62** | **12.82** | **24.53** | **41.15** | **15.26** |

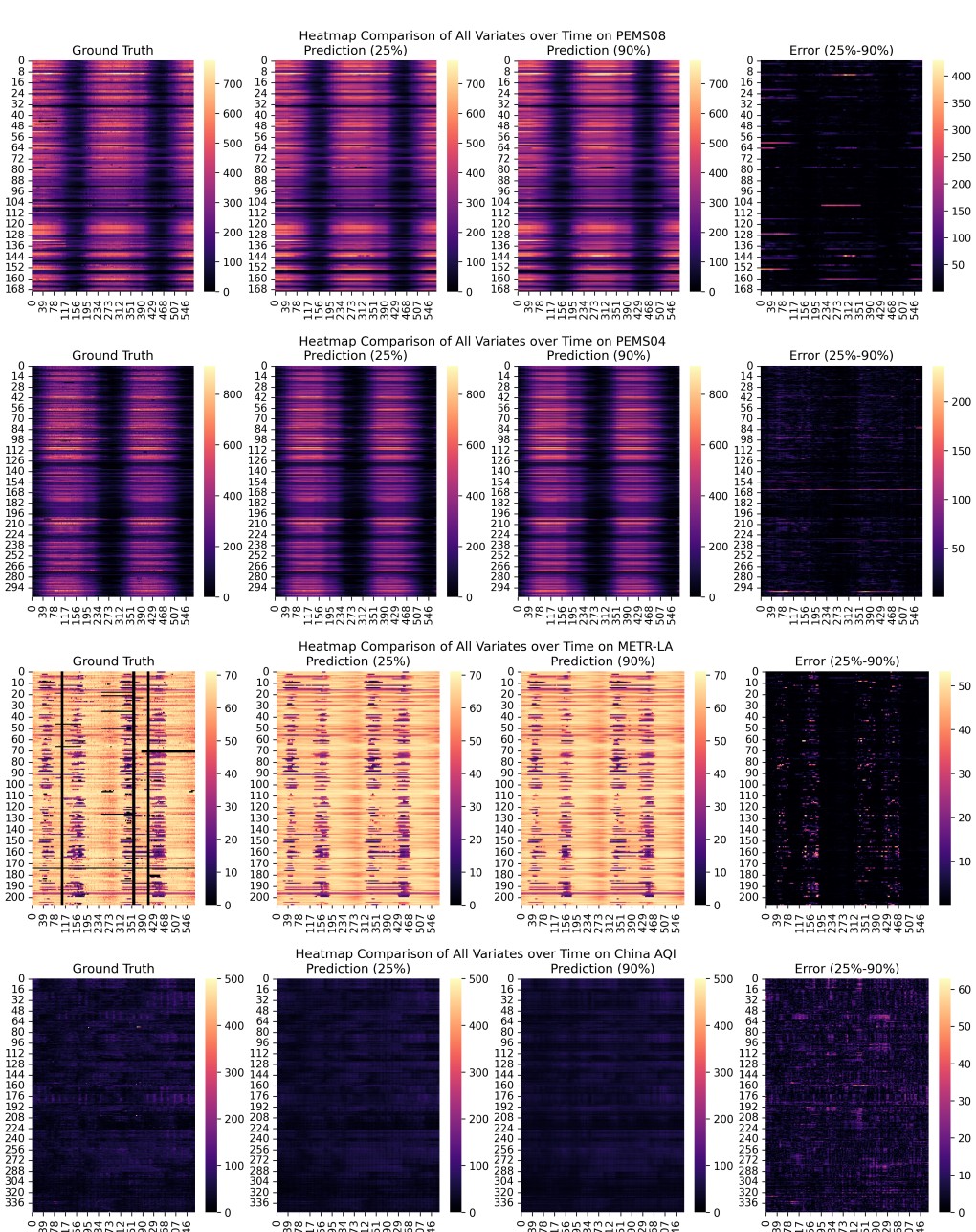

Figure 2: The forecasting heatmap produced by our VMPredictor on four datasets.

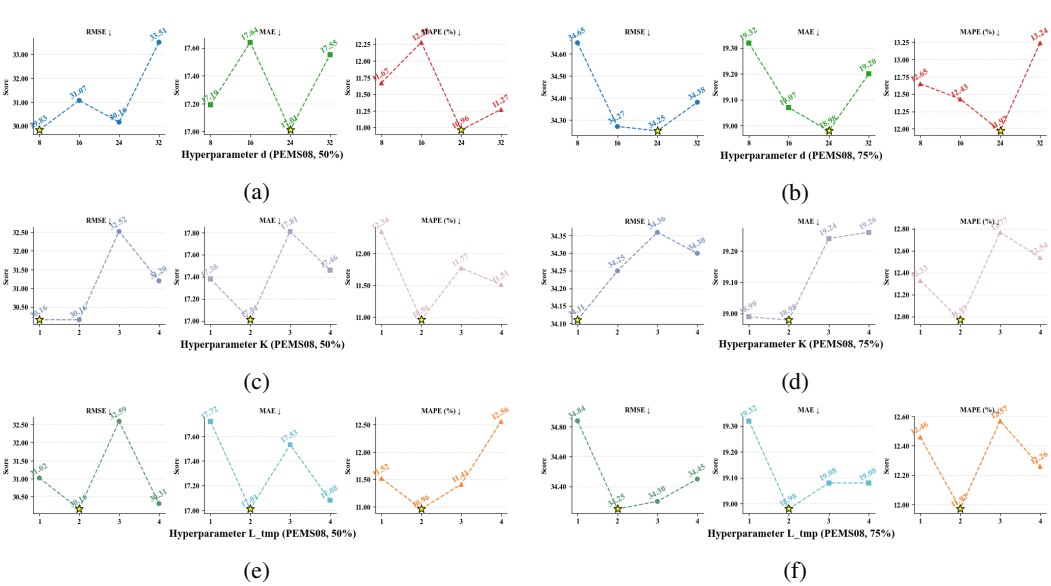

Figure 3: The hyperparameter sensitivity result on PEMS08.

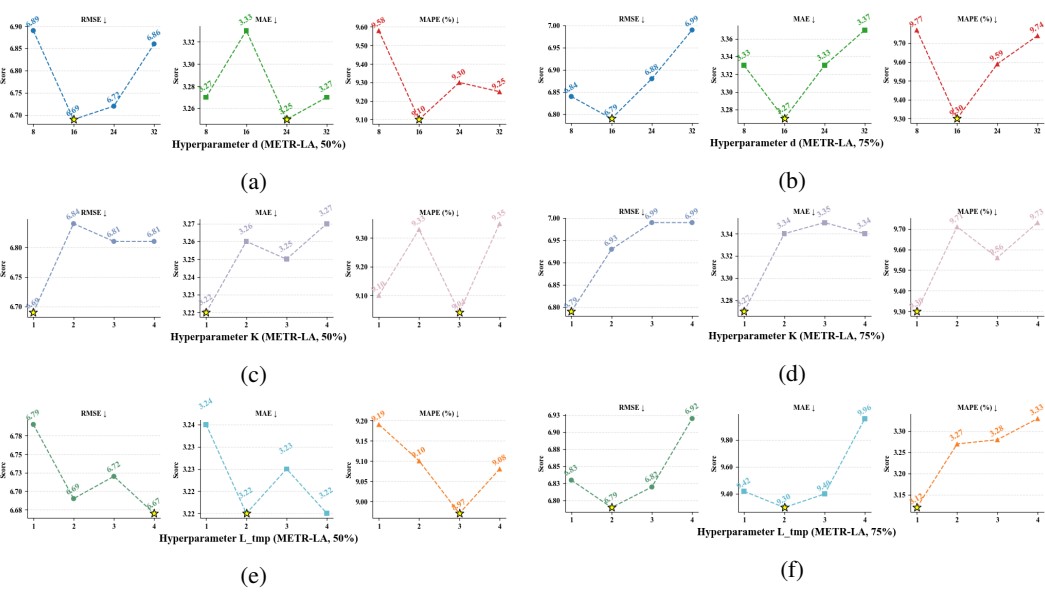

Figure 4: The hyperparameter sensitivity result on METR-LA.

Table 8: The model complexity of different methods on three datasets.

| Datasets | PEMS04(BSize=4) | | | PEMS08(BSize=16) | | | METR-LA(BSize=8) | | |
|---|---|---|---|---|---|---|---|---|---|
| | Memory | #Param | s/epo | Memory | #Param | s/epo | Memory | #Param | s/epo |
| BiTGraph | 0.64G | 61M | 110s | 0.73G | 39M | 27s | 0.66G | 45M | 129s |
| GinAR | 16.97G | 62M | 696s | 20.89G | 21M | 155s | 15.47G | 30M | 787s |
| VMPredictor | 3.95G | 146M | 155s | 3.66G | 136M | 77s | 2.03G | 87M | 377s |

## G   THE USE OF LARGE LANGUAGE MODELS (LLMS)

During the writing process of this article, the Large Language Model ChatGPT-5 was used mainly for relevant language polishing, for example, the Introduction and Related Work sections.

