# OpenReview forum: "Adaptive Graph Convolutional Network with Attention Fusion for Multivariate Time Series Forecasting with Variable Missing"
_ICLR.cc/2026/Conference — Submitted to ICLR 2026_

### Official Review · Reviewer_Fmme · 2025-10-27

**Soundness:** 3
**Presentation:** 3
**Contribution:** 3
**Rating:** 6
**Confidence:** 5

**Summary:**

This paper proposes VMPredictor, a novel end-to-end framework for multivariate time series forecasting (MTSF) under variable-missing scenarios. Unlike conventional two-stage imputation–forecasting methods, VMPredictor directly learns to model incomplete variables through adaptive representation learning. The framework integrates: (1) Adaptive Missing Filling and Enhancement Layer (AMFE Layer). (2) Embedding Layer: injects temporal and spatial embeddings. (3) Spatiotemporal Dependency Mining Layer (STDMLayer. (4) Multi-Head Temporal Self-Attention Layer (MHTSA): captures global temporal context for final prediction.

Comprehensive experiments on five real-world datasets (PEMS04/08, METR-LA, PEMS-BAY, China AQI) show that VMPredictor consistently outperforms 10+ SOTA baselines, especially at high missing rates (75%–90%).

**Strengths:**

1. The introduction of learnable missing embeddings allows dynamic representation of incomplete data, effectively reducing bias from fixed-value imputation. This design significantly improves robustness against missing patterns.
2. The combination of dynamic graph convolution and layer-normalized GRU enables simultaneous learning of temporal dynamics and spatial correlations. This achieves more precise inter-variable modeling compared to static GCN or vanilla GRU baselines.
3. The multi-head temporal self-attention layer captures long-range dependencies, while the SE-based enhancement layer adaptively reweights important channels and timestamps, leading to balanced local-global feature fusion.
4. The proposed method demonstrates consistent SOTA performance across diverse domains (traffic and air quality) and under severe missing conditions. Ablation and sensitivity analyses further confirm the effectiveness and interpretability of the design.

**Weaknesses:**

1. The introduction is somewhat disjointed. The authors could provide a deeper analysis of the core challenges in multivariate time series forecasting with variable missing data before introducing how the proposed method addresses these challenges.
2. The core technical innovation seems to lie mainly in the embedding layer, while the rest of the framework is similar to mainstream STGNNs. The authors are encouraged to further emphasize how the proposed embedding layer specifically mitigates existing problems, such as error accumulation in current methods.
3. The ablation experiment can consider directly removing all the additional embeddings. This will further demonstrate the effectiveness of the core technical contribution of this paper and its significance in this task.
4. It would be beneficial to conduct additional experiments in more widely used missing-data scenarios [1] to further verify the robustness of VMPredictor.
5. There are some formatting errors, such as the “Table ??” on page 13, line 684, and page 17, line 888.

[1] Graph-based Forecasting with Missing Data through Spatiotemporal Downsampling

**Questions:**

1. I’m curious whether the proposed model can be transferred to random missing or block missing tasks.
2. Is the superior performance of the proposed method mainly attributed to the model architecture or to the specifically designed embedding layer?
3. If the proposed embedding method is transferred to other backbones, will it be able to improve their performance?

---

> ### Author Response · Authors · 2025-12-01
> **Response to Reviewer Fmme**
>
> **W1:** The introduction is somewhat disjointed. The authors could provide a deeper analysis of the core challenges in multivariate time series forecasting with variable missing data before introducing how the proposed method addresses these challenges.
>
> **Res1:** Thank you very much for your constructive suggestion. We fully agree that the introduction would benefit from a clearer and more coherent discussion of the core challenges in multivariate time-series forecasting with variable missing data. In the revised manuscript, we have substantially improved this part by adding a more systematic analysis of the key difficulties.
>
> **W2:** The core technical innovation seems to lie mainly in the embedding layer, while the rest of the framework is similar to mainstream STGNNs. The authors are encouraged to further emphasize how the proposed embedding layer specifically mitigates existing problems, such as error accumulation in current methods.
>
> **Res2:** Thank you very much for your insightful comment. We appreciate the opportunity to clarify the technical contributions of our work. Although the overall architecture follows the common paradigm adopted in many STGNN-based forecasting models, the proposed embedding layer is not merely an auxiliary component but plays a central role in addressing a key limitation of existing methods—error accumulation caused by improper handling of missing values. Specifically, conventional STGNNs generally rely on predefined imputation strategies (e.g., zero-filling, mean-filling, or last-value carrying) before feeding the data into the model. These predefined strategies inevitably distort the original data distribution, especially under high missing rates, causing early-stage imputation errors to propagate through subsequent layers and accumulate over time. In contrast, our adaptive missing-value completion embedding replaces missing entries with learnable embeddings, which ***(1) preserve the distributional characteristics of observed data, (2) avoid injecting biased artificial values, and (3) allow the model to update the missing-value representation jointly with forecasting objectives.*** Furthermore, the embedding layer is tightly integrated with the temporal and channel-wise squeezing–excitation mechanisms, allowing the model to emphasize informative temporal slices and feature dimensions while suppressing uninformative or noisy ones. This design significantly reduces the cascading effect of early imputation noise, thereby mitigating error accumulation throughout the network.

---

> ### Author Response · Authors · 2025-12-01
> **Response to Reviewer Fmme**
>
> **W3:** The ablation experiment can consider directly removing all the additional embeddings. This will further demonstrate the effectiveness of the core technical contribution of this paper and its significance in this task.
>
> **Res3:**  Thank you very much for your valuable suggestion. In response, we have added an additional ablation experiment in which all auxiliary embedding layers are removed to more thoroughly assess their contribution. The updated results are presented in the revised ablation table. As shown, the performance drops as removing all embeddings. It is worth noting that,  when we remove all embedding variables $E_p$, $E_X$ and $E_a$, we find that at low missing rates, removing all embedding variables causes a faster decline in model performance than removing $E_p$ while at high missing rates, the opposite is true. This is because when the observed data is sufficient, the model relies more on structural representations $E_X$ and $E_a$, while when observation data is insufficient, structural representations implicitly introduce noise. This indirectly reflects the importance of various embeddings.
>
> ### Table1: The result of ablation studies on PEMS04 and PEMS-BAY under the missing rates of 25\% and 90\%.
>
> #### r = 25%
> | Method        | MAE | RMSE | MAPE | MAE | RMSE | MAPE |
> |---------------|-----|------|------|-----|------|------|
> |               | **PEMS04** | | | **PEMS-BAY** | | |
> | zero pefill      | 20.49 | 33.98 | 14.32 | 1.78 | **3.96** | 4.01 |
> | w/o. IE         | 19.86 | 33.48 | 13.55 | 1.81 | 4.09 | 4.16 |
> | w/o. $E_p$         | 21.34 | 34.47 | 14.75 | 1.83 | 4.11 | 4.11 |
> | w/o. $A_s$         | 19.92 | 33.37 | 14.05 | 1.81 | 4.07 | 4.11 |
> | w/o. $A_d$         | 20.00 | 33.35 | **13.44** | 1.80 | 3.99 | 4.06 |
> | w/o. MHTSA         | 20.96 | 34.44 | 15.39 | 1.80 | 3.99 | 4.06 |
> | w/o. EM         | 21.23 | 34.53 | 14.83 | 1.84 | 4.14 | 4.35 |
> | **VMPredictor** | **19.75** | **33.34** | 13.72 | **1.77** | 3.99 | **4.00** |
>
> #### r = 90%
> | Method        | MAE | RMSE | MAPE | MAE | RMSE | MAPE |
> |---------------|-----|------|------|-----|------|------|
> |               | **PEMS04** | | | **PEMS-BAY** | | |
> | zero pefill      | **21.90** | 36.84 | 15.89 | 2.34 | 5.14 | 5.34 |
> | w/o. IE         | 21.95 | **36.40** | 16.03 | 2.41 | 5.11 | 5.52 |
> | w/o. $E_p$         | 25.82 | 40.10 | 18.71 | 2.47 | 5.29 | 5.77 |
> | w/o. $A_s$         | 23.19 | 37.43 | 17.14 | 2.30 | 5.00 | 5.23 |
> | w/o. $A_d$         | 23.51 | 38.15 | 18.59 | 2.42 | 5.15 | 5.50 |
> | w/o. MHTSA         | 23.05 | 37.43 | 17.14 | 2.46 | 5.17 | 5.62 |
> | w/o. EM         | 23.63 | 38.07 | 16.93 | 2.46 | 5.15 | 5.63 |
> | **VMPredictor** | 22.18 | 36.81 | **15.61** | **2.28** | **4.96** | **5.20** |

---

> ### Author Response · Authors · 2025-12-01
> **Response to Reviewer Fmme**
>
> **W4:** It would be beneficial to conduct additional experiments in more widely used missing-data scenarios [1] to further verify the robustness of VMPredictor.
>
> **Res4:** Thank you very much for your insightful comment and valuable suggestion. Following your suggestion, we evaluated the prediction capability of VMPredictor under both random missing and block missing scenarios on the PEMS04 and PEMS08 datasets. The corresponding results are shown in the tables. As observed, VMPredictor consistently achieves the best performance, further demonstrating its superior effectiveness in handling missing-data prediction tasks.
> ### Table1: The forecasting performance of different methods on PEMS04 under RM and BM.
>
> #### r = 25%
> | Method        | MAE | RMSE | MAPE | MAE | RMSE | MAPE |
> |---------------|-----|------|------|-----|------|------|
> |               | **PEMS04 (RM)** | | | **PEMS04 (BM)** | | |
> | BiTGraph      | 19.47 | 31.61 | **13.39** | 21.41 | **33.98** | 14.97 |
> | GinAR         | 20.24 | 33.21 | 15.03 | 23.61 | 36.42 | 16.27 |
> | **VMPredictor** | **19.43** | **30.98** | 13.64 | **20.35** | 34.38 | **13.60** |
>
> #### r = 50%
> | Method        | MAE | RMSE | MAPE | MAE | RMSE | MAPE |
> |---------------|-----|------|------|-----|------|------|
> |               | **PEMS04 (RM)** | | | **PEMS04 (BM)** | | |
> | BiTGraph      | 20.48 | 32.49 | 14.55 | 24.02 | 37.49 | 16.62 |
> | GinAR         | 21.37 | 35.53 | 16.22 | 25.77 | 38.22 | 18.04 |
> | **VMPredictor** | **19.77** | **31.75** | **13.79** | **23.05** | **36.50** | **15.56** |
>
> #### r = 75%
> | Method        | MAE | RMSE | MAPE | MAE | RMSE | MAPE |
> |---------------|-----|------|------|-----|------|------|
> |               | **PEMS04 (RM)** | | | **PEMS04 (BM)** | | |
> | BiTGraph      | 21.53 | 34.49 | **14.58** | 25.95 | 40.09 | 17.66 |
> | GinAR         | 24.72 | 38.42 | 18.77 | 26.88 | 41.73 | 19.41 |
> | **VMPredictor** | **21.49** | **33.69** | 15.08 | **24.26** | **38.17** | **17.05** |
>
> #### r = 90%
> | Method        | MAE | RMSE | MAPE | MAE | RMSE | MAPE |
> |---------------|-----|------|------|-----|------|------|
> |               | **PEMS04 (RM)** | | | **PEMS04 (BM)** | | |
> | BiTGraph      | 24.13 | 37.71 | 16.28 | 27.29 | 41.91 | 18.38 |
> | GinAR         | 26.28 | 40.01 | 20.01 | 28.08 | 43.51 | 21.33 |
> | **VMPredictor** | **22.90** | **36.54** | **15.82** | **25.37** | **39.07** | **17.63** |
>
> ### Table2: The forecasting performance of different methods on PEMS08 under RM and BM.
>
> #### r = 25%
> | Method        | MAE | RMSE | MAPE | MAE | RMSE | MAPE |
> |---------------|-----|------|------|-----|------|------|
> |               | **PEMS08 (RM)** | | | **PEMS08 (BM)** | | |
> | BiTGraph      | 16.29 | 25.54 | 10.22 | 18.81 | 30.25 | 11.94 |
> | GinAR         | 20.28 | 32.24 | 14.61 | 21.51 | 33.08 | 21.51 |
> | **VMPredictor** | **15.25** | **24.52** | 9.91 | **17.35** | 28.39 | **11.33** |
>
> #### r = 50%
> | Method        | MAE | RMSE | MAPE | MAE | RMSE | MAPE |
> |---------------|-----|------|------|-----|------|------|
> |               | **PEMS08 (RM)** | | | **PEMS08 (BM)** | | |
> | BiTGraph      | 16.88 | 26.30 | **10.82** | 22.74 | 37.07 | 13.83 |
> | GinAR         | 21.74 | 35.07 | 16.76 | 25.96 | 40.07 | 19.83 |
> | **VMPredictor** | **16.14** | **25.31** | 10.91 | **20.35** | **34.38** | **12.60** |
>
> #### r = 75%
> | Method        | MAE | RMSE | MAPE | MAE | RMSE | MAPE |
> |---------------|-----|------|------|-----|------|------|
> |               | **PEMS08 (RM)** | | | **PEMS08 (BM)** | | |
> | BiTGraph      | 18.40 | 28.82 | 12.13 | 26.83 | 42.84 | 15.99 |
> | GinAR         | 24.39 | 39.53 | 18.91 | 29.57 | 46.18 | 19.92 |
> | **VMPredictor** | **17.24** | **27.52** | **11.60** | **23.37** | **39.07** | **14.63** |
>
> #### r = 90%
> | Method        | MAE | RMSE | MAPE | MAE | RMSE | MAPE |
> |---------------|-----|------|------|-----|------|------|
> |               | **PEMS08 (RM)** | | | **PEMS08 (BM)** | | |
> | BiTGraph      | 21.55 | 33.94 | 14.36 | 30.08 | 46.56 | 18.08 |
> | GinAR         | 26.21 | 42.38 | 20.14 | 29.69 | 46.03 | 21.39 |
> | **VMPredictor** | **19.72** | **32.62** | **12.82** | **24.53** | **41.15** | **15.26** |
>
> **W5:** There are some formatting errors, such as the “Table ??” on page 13, line 684, and page 17, line 888.
>
> **Res5:** We sincerely appreciate your careful reading of our manuscript and for pointing out the typographical errors. We have thoroughly rechecked the entire paper and corrected all relevant writing and formatting issues.

---

> ### Author Response · Authors · 2025-12-01
> **Response to Reviewer Fmme**
>
> **Q1:** I’m curious whether the proposed model can be transferred to random missing or block missing tasks.
>
> **ResQ1:** Thank you very much for your insightful question. We would like to clarify that the proposed VMPredictor is not limited to a specific missing-data scenario. In fact, the model can be directly extended to a variety of missing-data prediction tasks, including both random missing patterns and block-wise missing patterns. This generalizability stems from the adaptive missing-value completion mechanism and the dynamic spatiotemporal modeling design, which together enable VMPredictor to effectively handle diverse and complex missing structures. We evaluated the prediction capability of VMPredictor under both random missing and block missing scenarios on the PEMS04 and PEMS08 datasets. The corresponding results are shown in the above tables . As observed, VMPredictor consistently achieves the best performance, further demonstrating its superior effectiveness in handling missing-data prediction tasks.
>
> **Q2:** Is the superior performance of the proposed method mainly attributed to the model architecture or to the specifically designed embedding layer?
>
> **ResQ2:** Thank you for your valuable question. The superior performance of the proposed VMPredictor does not come solely from the embedding layer nor solely from the model architecture, but rather from the synergistic effect between the two.On the one hand, the specifically designed adaptive embedding layer plays a crucial role in handling variable missing data. By replacing missing entries with learnable representations instead of predefined constants, it effectively mitigates distribution distortion and prevents error accumulation, which are common limitations in existing methods. On the other hand, the overall model architecture—including the dynamic spatiotemporal convolutional recurrent units and the temporal–channel excitation mechanism—further enhances the model’s ability to capture heterogeneous temporal dependencies and dynamic spatial correlations. Therefore, the performance improvement originates from the combination of the adaptive embedding strategy and the enhanced spatiotemporal modeling architecture. Ablation studies in the paper also confirm that both components contribute significantly and jointly lead to the strong performance of VMPredictor.
>
> **Q3:** If the proposed embedding method is transferred to other backbones, will it be able to improve their performance?
>
> **ResQ3:** Thank you for your valuable question. We applied the proposed ADAPTIVE MISSING FILLING ENHANCEMENT LAYER (AMFELayer) to the BiTGraph model (because AMFELayer requires sequence parallel processing capabilities, making BiTGraph a suitable choice), and then tested its performance on the PEMS04 and PEMS08 datasets in the random missing data scenario. The experimental results are shown in the table. It is easy to see that the AMFELayer improves the prediction performance of BiTGraph. We also speculate that the AMFELayer may also improve the performance of Transformer-based models, and we will continue to test this in the future.
>
> ### Table1: AMFELayer Performance Test on PEMS04 and PEMS08 under RM.
>
> #### r = 25%
> | Method        | MAE | RMSE | MAPE | MAE | RMSE | MAPE |
> |---------------|-----|------|------|-----|------|------|
> |               | **PEMS04 (RM)** | | | **PEMS08 (RM)** | | |
> | BiTGraph      | **19.47** | 31.61 | **13.39** | 16.29 | 25.54 | **10.22** |
> | +AMFELayer         | **19.47** | **31.28** | 13.88 | **16.11** | **25.48** | 10.48 |
>
> #### r = 50%
> | Method        | MAE | RMSE | MAPE | MAE | RMSE | MAPE |
> |---------------|-----|------|------|-----|------|------|
> |               | **PEMS04 (RM)** | | | **PEMS08 (RM)** | | |
> | BiTGraph      | 20.48 | **32.49** | 14.55 | 16.88 | 26.30 | 10.82 |
> | +AMFELayer         | **20.28** | 32.54 | **14.04** | **16.78** | **26.27** | **10.77** |
>
> #### r = 75%
> | Method        | MAE | RMSE | MAPE | MAE | RMSE | MAPE |
> |---------------|-----|------|------|-----|------|------|
> |               | **PEMS04 (RM)** | | | **PEMS08 (RM)** | | |
> | BiTGraph      | **21.53** | 34.49 | **14.58** | **18.40** | **28.82** | 12.13 |
> | +AMFELayer         | 21.66 | **34.21** | 15.07 | 18.51 | 28.96 | **11.73** |
>
> #### r = 90%
> | Method        | MAE | RMSE | MAPE | MAE | RMSE | MAPE |
> |---------------|-----|------|------|-----|------|------|
> |               | **PEMS04 (RM)** | | | **PEMS04 (BM)** | | |
> | BiTGraph      | 24.13 | 37.71 | **16.28** | 21.56 | 33.94 | 14.36 |
> | +AMFELayer         | **24.08** | 37.78 | **16.28** | **21.47** | **33.88** | **13.53** |

---

### Official Review · Reviewer_PhTa · 2025-10-31

[review text omitted: it was posted to a different submission]

---

> ### Comment · Reviewer_PhTa · 2025-11-28
> **Correct Review**
>
> I sincerely apologize for the confusion caused to the authors. My previous review comments were indeed not aligned with the content of this paper. I am not sure whether this discrepancy resulted from an error on my part or from a system issue.
>
> Below are the correct review comments. Since I can no longer modify the scores in the system, I have placed my ratings at the bottom in the largest font size for clarity.
>
> ### Summary
>
> This paper presents the VMPredictor model, designed to tackle forecasting challenges arising from missing data in multivariate time series prediction. The model integrates adaptive missing-value imputation with graph convolutional networks, effectively capturing both spatial and temporal dependencies. It performs particularly well in scenarios with severe data loss.
>
> ### Strengths
>
> 1. The paper introduces VMPredictor, a novel approach for multivariate time series forecasting under missing-data conditions.
> 2. The proposed framework captures spatial and temporal dependencies through dynamic graph convolution and Gated Recurrent Units (GRUs). This integrated spatiotemporal modeling demonstrates strong robustness, especially when missing-data rates are high.
> 3. Extensive experiments on five real-world datasets show that VMPredictor consistently outperforms ten baseline models, maintaining superior performance even under severe missing-data scenarios.
> 4. The authors provide the source code, ensuring reproducibility of the model.
>
> ### Weaknesses
>
> 1. Random missing and block missing are also widely studied tasks [1]. The authors may consider analyzing VMPredictor’s performance under these settings in the supplementary materials.
> 2. A more in-depth discussion of the limitations of the end-to-end model is needed. In particular, the authors state that the model cannot flexibly handle missing values, so they should clarify the issues that arise when directly filling missing entries with zeros or other naive methods. This would better highlight the benefits of the Adaptive Missing Filling Enhancement Layer.
> 3. There are several grammatical and formatting issues, such as those found on line 353 of page 7 and line 684 of page 13.
>
> [1] Xiaodan Chen, Xiucheng Li, Bo Liu, and Zhijun Li. *Biased temporal convolution graph network for time series forecasting with missing values*. In *The Twelfth International Conference on Learning Representations*, 2023.
>
> # Soundness: 3
>
> # Presentation: 3
>
> # Contribution: 3
>
> # Confidence: 4
>
> # Rating: 6

---

> ### Author Response · Authors · 2025-12-01
> **Response to Reviewer PhTa**
>
> **W1:** Random missing and block missing are also widely studied tasks [1]. The authors may consider analyzing VMPredictor’s performance under these settings in the supplementary materials.
>
> **Res1:** Thank you very much for your insightful comment and valuable suggestion. Following your suggestion, we evaluated the prediction capability of VMPredictor under both random missing and block missing scenarios on the PEMS04 and PEMS08 datasets. The corresponding results are shown in the tables. As observed, VMPredictor consistently achieves the best performance, further demonstrating its superior effectiveness in handling missing-data prediction tasks.
> ### Table1: The forecasting performance of different methods on PEMS04 under RM and BM.
>
> #### r = 25%
> | Method        | MAE | RMSE | MAPE | MAE | RMSE | MAPE |
> |---------------|-----|------|------|-----|------|------|
> |               | **PEMS04 (RM)** | | | **PEMS04 (BM)** | | |
> | BiTGraph      | 19.47 | 31.61 | **13.39** | 21.41 | **33.98** | 14.97 |
> | GinAR         | 20.24 | 33.21 | 15.03 | 23.61 | 36.42 | 16.27 |
> | **VMPredictor** | **19.43** | **30.98** | 13.64 | **20.35** | 34.38 | **13.60** |
>
> #### r = 50%
> | Method        | MAE | RMSE | MAPE | MAE | RMSE | MAPE |
> |---------------|-----|------|------|-----|------|------|
> |               | **PEMS04 (RM)** | | | **PEMS04 (BM)** | | |
> | BiTGraph      | 20.48 | 32.49 | 14.55 | 24.02 | 37.49 | 16.62 |
> | GinAR         | 21.37 | 35.53 | 16.22 | 25.77 | 38.22 | 18.04 |
> | **VMPredictor** | **19.77** | **31.75** | **13.79** | **23.05** | **36.50** | **15.56** |
>
> #### r = 75%
> | Method        | MAE | RMSE | MAPE | MAE | RMSE | MAPE |
> |---------------|-----|------|------|-----|------|------|
> |               | **PEMS04 (RM)** | | | **PEMS04 (BM)** | | |
> | BiTGraph      | 21.53 | 34.49 | **14.58** | 25.95 | 40.09 | 17.66 |
> | GinAR         | 24.72 | 38.42 | 18.77 | 26.88 | 41.73 | 19.41 |
> | **VMPredictor** | **21.49** | **33.69** | 15.08 | **24.26** | **38.17** | **17.05** |
>
> #### r = 90%
> | Method        | MAE | RMSE | MAPE | MAE | RMSE | MAPE |
> |---------------|-----|------|------|-----|------|------|
> |               | **PEMS04 (RM)** | | | **PEMS04 (BM)** | | |
> | BiTGraph      | 24.13 | 37.71 | 16.28 | 27.29 | 41.91 | 18.38 |
> | GinAR         | 26.28 | 40.01 | 20.01 | 28.08 | 43.51 | 21.33 |
> | **VMPredictor** | **22.90** | **36.54** | **15.82** | **25.37** | **39.07** | **17.63** |
>
> ### Table2: The forecasting performance of different methods on PEMS08 under RM and BM.
>
> #### r = 25%
> | Method        | MAE | RMSE | MAPE | MAE | RMSE | MAPE |
> |---------------|-----|------|------|-----|------|------|
> |               | **PEMS08 (RM)** | | | **PEMS08 (BM)** | | |
> | BiTGraph      | 16.29 | 25.54 | 10.22 | 18.81 | 30.25 | 11.94 |
> | GinAR         | 20.28 | 32.24 | 14.61 | 21.51 | 33.08 | 21.51 |
> | **VMPredictor** | **15.25** | **24.52** | 9.91 | **17.35** | 28.39 | **11.33** |
>
> #### r = 50%
> | Method        | MAE | RMSE | MAPE | MAE | RMSE | MAPE |
> |---------------|-----|------|------|-----|------|------|
> |               | **PEMS08 (RM)** | | | **PEMS08 (BM)** | | |
> | BiTGraph      | 16.88 | 26.30 | **10.82** | 22.74 | 37.07 | 13.83 |
> | GinAR         | 21.74 | 35.07 | 16.76 | 25.96 | 40.07 | 19.83 |
> | **VMPredictor** | **16.14** | **25.31** | 10.91 | **20.35** | **34.38** | **12.60** |
>
> #### r = 75%
> | Method        | MAE | RMSE | MAPE | MAE | RMSE | MAPE |
> |---------------|-----|------|------|-----|------|------|
> |               | **PEMS08 (RM)** | | | **PEMS08 (BM)** | | |
> | BiTGraph      | 18.40 | 28.82 | 12.13 | 26.83 | 42.84 | 15.99 |
> | GinAR         | 24.39 | 39.53 | 18.91 | 29.57 | 46.18 | 19.92 |
> | **VMPredictor** | **17.24** | **27.52** | **11.60** | **23.37** | **39.07** | **14.63** |
>
> #### r = 90%
> | Method        | MAE | RMSE | MAPE | MAE | RMSE | MAPE |
> |---------------|-----|------|------|-----|------|------|
> |               | **PEMS08 (RM)** | | | **PEMS08 (BM)** | | |
> | BiTGraph      | 21.55 | 33.94 | 14.36 | 30.08 | 46.56 | 18.08 |
> | GinAR         | 26.21 | 42.38 | 20.14 | 29.69 | 46.03 | 21.39 |
> | **VMPredictor** | **19.72** | **32.62** | **12.82** | **24.53** | **41.15** | **15.26** |

---

> > ### Author Response · Authors · 2025-12-01
> > **Response to Reviewer PhTa**
> >
> > **W2:** A more in-depth discussion of the limitations of the end-to-end model is needed. In particular, the authors state that the model cannot flexibly handle missing values, so they should clarify the issues that arise when directly filling missing entries with zeros or other naive methods. This would better highlight the benefits of the Adaptive Missing Filling Enhancement Layer.
> >
> > **Res2:** Thank you very much for your valuable suggestion and question. We have provided a more in-depth discussion on the limitations of end-to-end models. For recent mainstream models, missing values are typically handled by predefined imputation methods (e.g., filling with 0, the last observed value, or the mean). While intuitive, this approach can reduce model flexibility and may alter the original data distribution, especially when the missing rate is high. For instance, if missing values are filled with 0 and the missing rate exceeds 50%, the resulting data distribution can significantly deviate from the original, which in turn directly affects model prediction performance. This issue is particularly pronounced under such circumstances. In contrast, our proposed adaptive missing value imputation enhancement layer effectively mitigates this problem. By using trainable embeddings to fill missing values, the model’s flexibility is increased while the risk of data distribution shift is reduced.
> >
> > **W3:** There are several grammatical and formatting issues, such as those found on line 353 of page 7 and line 684 of page 13.
> >
> > **Res3:** We sincerely appreciate your careful reading of our manuscript and for pointing out the typographical errors. We have thoroughly rechecked the entire paper and corrected all relevant writing and formatting issues.

---

### Official Review · Reviewer_wwt7 · 2025-11-01

**Soundness:** 2
**Presentation:** 2
**Contribution:** 2
**Rating:** 2
**Confidence:** 4

**Summary:**

This paper attempts to resolve the challenge of variable incompleteness in time series forecasting by proposing VMPredictor, a model that effectively captures spatiotemporal dependencies among incomplete variables to improve forecasting accuracy. Specifically, VMPredictor incorporates two key modules: the Adaptive Missing Filling and Enhancement Layer, which imputes and refines incomplete variables, and the Spatiotemporal Dependency Mining Layer, which captures both intra- and inter-series dependencies. Extensive experiments on five benchmark datasets demonstrate that the proposed model achieves state-of-the-art performance.

**Strengths:**

- This paper proposes a one-stage framework for time series forecasting with variable missing.
- It introduces a learnable embedding to alleviate learning bias caused by fixed fill-in values.
- Experimental results demonstrate that the proposed model achieves superior performance, even at high missing rates.

**Weaknesses:**

- This paper aims to address the problem of variable missing in time series forecasting. However, the proposed method lacks specific mechanisms or designs that explicitly target this issue, leading to a misalignment between the stated motivation and the model design. The author should clarify the rationale for introducing the squeeze-excitation module and the Dynamic Graph Convolution Layer-Normalized Gated Recurrent Unit, and explain how these components contribute to handling variable missing problems. Moreover, the proposed model primarily integrates existing modules, and its overall novelty appears limited. The author should more clearly highlight the unique contributions or theoretical insights beyond this integration.

-  Several arguments require clarification. For example, how does the learnable embedding $E_X \in \mathbb{R}^{T\times N\times d}$ alleviate parameter bias? How is $A_s$ defined for datasets that lack a predefined graph structure? In Eq. 14, the definitions of $\alpha$, $\beta$, and $\gamma$ are unclear. In line 271, the author states that 'where $W_v$, $W_n$, $b_v$, and $b_n$ are all trainable parameters', but these variables are not introduced or defined earlier in the paper.

- The hyperparameter settings require clarification. Specifically, the values of $d$, $d_s$, and $K$ vary across different datasets. Could the authors explain the rationale behind these choices and provide guidance on how to select appropriate values for new datasets?

**Questions:**

- There are numerous grammatical and stylistic errors in this paper. For example, the forecasting window size is inconsistently denoted as $F$, $H$, and $T$ in lines 157, 161, 315, and 316.  In Eq. 4, subscript notations are used inconsistently (1:T and 1:t). In addition, there are several typographical and capitalization errors, such as forecastinh (line 16), ': ,' (line 353), ', ,' (line 761), and 'Table ??' (lines 685 and 889). These issues significantly affect the readability and professionalism of the paper.

---

> ### Author Response · Authors · 2025-11-30
> **Response to Reviewer wwt7**
>
> **W1:** This paper aims to address the problem of variable missing in time series forecasting. However, the proposed method lacks specific mechanisms or designs that explicitly target this issue, leading to a misalignment between the stated motivation and the model design. The author should clarify the rationale for introducing the squeeze-excitation module and the Dynamic Graph Convolution Layer-Normalized Gated Recurrent Unit, and explain how these components contribute to handling variable missing problems. Moreover, the proposed model primarily integrates existing modules, and its overall novelty appears limited. The author should more clearly highlight the unique contributions or theoretical insights beyond this integration.
>
> **Res1:** Thank you very much for your valuable question. To address the issue of variable missingness in time-series forecasting, we have considered the following aspects: **1)  Handling variable missingness during the prediction stage:** ariable missingness refers to the situation in which the observations of certain nodes are entirely absent at a specific time step t. Unlike BiTGraph and GinAR, which directly process the incomplete data, our method introduces an *adaptive padding mechanism*. Specifically, we design a learnable feature embedding $E_X$ and combine it with the original data through the masking matrix $M$. The non-missing entries retain their original values, while the missing entries are replaced with learnable parameters. This strategy avoids injecting noise or bias into the model and helps it more effectively cope with the missing-value problem. **2) Incorporating a squeeze-and-excitation (SE) module:** The SE module is introduced to better extract both temporal characteristics and channel-wise dependencies in the sequence data. It contains two components: temporal excitation and channel excitation. On the temporal side, real-world time series inevitably contain noise, and only certain critical time points within a time slice contribute significantly to prediction accuracy. The temporal SE component enables the model to attend more to these important time points. On the channel side, different feature channels have varying levels of relevance—some are beneficial, while others may be redundant or noisy. The channel SE component helps the model highlight the most informative channels. **3) Design of the DGCLNGRU:** We propose a novel architecture that integrates dynamic graph convolution with a layer-normalized GRU, aiming to capture the spatiotemporal dependencies more effectively. Rather than simply combining GCN and GRU as done in prior works, our design further incorporates dropout and layer normalization to enhance robustness and training stability. This structural formulation has not been previously explored.
>
> **W2:** Several arguments require clarification. For example, how does the learnable embedding $E_X$ alleviate parameter bias? How is $A_s$ defined for datasets that lack a predefined graph structure? In Eq. 14, the definitions $\alpha$, $\beta$, and $\gamma$ are unclear. In line 271, the author states that 'where $W_v$, $W_n$, $b_v$, and $b_n$ are all trainable parameters', but these variables are not introduced or defined earlier in the paper.
>
> **Res2:** Thank you for your careful reading and insightful questions. We address them point by point below. 1) On $E_X$, $E_X$ is defined as a learnable embedding for missing entries. Concretely, positions corresponding to missing observations are replaced with the trainable embedding vectors from $E_X$, while observed positions remain unchanged. During training, these embeddings are updated jointly with the model parameters based on the prediction loss, which allows the model to learn a data-driven representation for missing values. This approach is more flexible than using fixed imputations (e.g., zeros or last-observed values) and avoids artificially altering the observed-data distribution. 2) On reliance on graph structure, VMPredictor is designed to exploit both prior (static) and data-driven (dynamic) topology information by using two graphs: a predefined static graph $A_s$ and a learned dynamic graph $A_d$. For datasets where a reliable predefined graph is not available, the model can operate using only the dynamic graph $A_d$ i.e., VMPredictor does not strictly depend on the availability of $A_s$. In other words, the model flexibly leverages whichever structural information is present. 3) On the hyperparameters $\alpha$, $\beta$, and $\gamma$, these scalar hyperparameters control the relative  weighting of information flowing through different computational graph paths in the architecture. They are used to balance contributions from multiple routes during feature aggregation.  4) On  $W_v$, $W_n$, $b_v$, and $b_n$, these symbols were typographical errors introduced during manuscript preparation. We apologize for the confusion and have corrected them in the revised manuscript.

---

> ### Author Response · Authors · 2025-11-30
> **Response to Reviewer wwt7**
>
> **W3:** The hyperparameter settings require clarification. Specifically, the values of $d$, $d_s$ and $K$ , vary across different datasets. Could the authors explain the rationale behind these choices and provide guidance on how to select appropriate values for new datasets?
>
> **Res3:** Thank you for raising this important question regarding the hyperparameter settings of $d$, $d_s$ and $K$. We clarify our design principles and provide practical guidelines for new datasets as follows.1) Rationale behind choosing $d$, $d_s$. The dimensions $d$ and $d_s$ determine the capacity of temporal feature extraction and structural embedding, respectively. In our experiments, we adapt these values to the scale and complexity of each dataset: For datasets with **larger spatial scales or richer temporal patterns** (e.g., PEMS04/China AQI), a moderately larger $d$ and $d_s$ helps capture more expressive patterns; For **smaller or less complex** datasets (e.g., METR-LA/PEMSBAY), we reduce these dimensions to avoid over-parameterization and ensure stable training. 2) Rationale for selecting $K$. The parameter $K$ controls how many hops of spatial information the model aggregates. In our experiments, we adapt these values to the scale and complexity of each dataset: For datasets with **denser sensor deployments or strong long-range spatial correlations** benefit from a slightly larger $K$; For datasets with **sparse or weak spatial dependencies** perform best with smaller $K$ .Thus, the variation of $K$ across datasets reflects differences in their inherent spatial correlation ranges. 3) Guidelines for new datasets. For practitioners applying our method to new datasets, we recommend: First, Start with a moderate setting, e.g. $d$=24, $d_s$=20 and $K$=2; Second, Scale d and $d_s$ according to dataset complexity (number of nodes, temporal variability, noise level); Third, Tune $K$ by analyzing spatial correlation decay (e.g., through adjacency distance statistics, correlation heatmaps, or variograms). In our experience, these hyperparameters are not overly sensitive, and a coarse search in a small range typically yields robust performance.
>
> **Q1:** There are numerous grammatical and stylistic errors in this paper. For example, the forecasting window size is inconsistently denoted as $F$, $H$ and $T$ in lines 157, 161, 315, and 316. In Eq. 4, subscript notations are used inconsistently (1:T and 1:t). In addition, there are several typographical and capitalization errors, such as forecastinh (line 16), ': ,' (line 353), ', ,' (line 761), and 'Table ??' (lines 685 and 889). These issues significantly affect the readability and professionalism of the paper.
>
> **Res2Q1** We sincerely appreciate your careful reading of our manuscript and for pointing out the typographical errors. We have thoroughly rechecked the entire paper and corrected all relevant writing and formatting issues.

---

### Official Review · Reviewer_rmYy · 2025-11-03

**Soundness:** 3
**Presentation:** 2
**Contribution:** 3
**Rating:** 4
**Confidence:** 4

**Summary:**

The authors propose VMPredictor, an end-to-end framework that addresses key challenges in missing subset variable forecasting, including severe error accumulation, the lack of flexible mechanisms for handling missing data, and the overreliance on local spatiotemporal correlations in existing methods.

**Strengths:**

S1. The adaptive missing-data imputation and enhancement layer introduces learnable embeddings to adaptively fill missing positions and dynamically refine incomplete representations during training.

S2. The spatiotemporal dependency mining layer is built upon a dynamic graph convolutional gated recurrent unit, where dynamic graph convolution adaptively reconstructs spatial correlations and replaces all fully connected layers in the GRU to capture synchronized spatiotemporal dependencies.

**Weaknesses:**

W1. The definition and role of $E_p$ in Equation (11) are unclear. The author should clarify its purpose, explain how it relates to $E_a$, and elaborate on their connections with $E_{day}$ and $E_{week}$.

W2. There are several typographical errors, such as “Table ??” in Line 889 and “forecastinh” in Line 015.

W3. The author randomly masks a fixed proportion of data; therefore, results from multiple random seeds should be reported to demonstrate the stability of the proposed method. Furthermore, sufficient implementation details should be provided to facilitate code reproducibility.

W4. The author does not specify the source of the “China AQI” dataset. While it appears to refer to air quality index data, which is dynamically computed based on multiple pollutant concentrations, the author should clarify which specific pollutant was used to ensure consistency with the characteristics of other datasets.

**Questions:**

See W1-4.

---

> ### Author Response · Authors · 2025-11-30
> **Response to Reviewer rmYy**
>
> **W1:** The definition and role of $E_p$ in Equation (11) are unclear. The author should clarify its purpose, explain how it relates to E_a and elaborate on their connections with $E_{day}$ and $E_{week}$.
>
> **Res1:** Thank you very much for your insightful comment. We would like to clarify the distinction between the two embeddings. Specifically,  $E_p=[E_{day}||{E_week}]$ denotes the temporal exogenous–variable embedding, which provides the model with periodic temporal cues (e.g., daily and weekly rhythms). It is parallel to $E_a$, and the two represent different types of embeddings. While $E_a$ is a learnable spatiotemporal embedding parameterized within the model, $E_p$ serves as an externally derived temporal-context embedding. Therefore, they play complementary but fundamentally different roles in encoding temporal information.
>
> **W2:** There are several typographical errors, such as “Table ??” in Line 889 and “forecastinh” in Line 015.
>
> **Res2:** Thank you very much for pointing out these typographical errors. We sincerely apologize for the oversight that led to these mistakes. We have carefully rechecked the entire manuscript and corrected all identified writing errors.
>
> **W3:** The author randomly masks a fixed proportion of data; therefore, results from multiple random seeds should be reported to demonstrate the stability of the proposed method. Furthermore, sufficient implementation details should be provided to facilitate code reproducibility.
>
> **Res3:** Thank you for your valuable comments. The reason why the experimental results in the current version are not presented in the form of mean and variance is that we aimed to maintain consistency with the results reported in the GinAR paper. To ensure reproducibility, we have publicly released the source code of our model through an anonymous GitHub repository.
>
> **W4:** The author does not specify the source of the “China AQI” dataset. While it appears to refer to air quality index data, which is dynamically computed based on multiple pollutant concentrations, the author should clarify which specific pollutant was used to ensure consistency with the characteristics of other datasets.
>
> **Res4:** Thank you again for your valuable comment and question. We apologize for the oversight in not explicitly providing the source of the dataset. The China AQI dataset (from GinAR) used in our experiments is obtained from *https://quotsoft.net/air/*.  This dataset dynamically calculates air quality based on the concentrations of multiple pollutants, primarily including PM2.5, PM10, SO₂, and NO₂, which was collected by hundreds of sensor nodes scattered throughout China and has spatiotemporal dependence characteristics.

---

### Author Response · Authors · 2025-12-02
**General Response to All Reviewers and Area Chair**

We thank all reviewers for their insightful and constructive feedback. We have carefully addressed all concerns and significantly strengthened our manuscript with extensive new experiments and clarifications.

**Below is a summary of our major updates of the paper:**
   1. **Clarification of Motivation and Novelty**

       $\circ$ We have reorganized the Introduction section of the paper to provide a deeper analysis of the core challenges.

   2. **Extensive Expansion of Missing Scenarios**

       $\circ$ **New Missing Scenarios:** We have added two different missing scenarios: **Random Missing (RM)** and **Block Missing (BM)**.

       $\circ$ **Results:** As shown in the updated **Tables 6 \& 7**, VMPredictor consistently achieves state-of-the-art performance across varying missing rates compared to BiTGraph and GinAR.

   3. **Manuscript Correction**

       $\circ$ We reviewed the entire paper and corrected the writing and expression errors.

We believe these revisions will effectively address reviewers' concerns about motivation and generalization.

---

### Meta-Review · Area_Chair_eygw · 2026-01-07

**Summary:**

This paper proposes VMPredictor, an end-to-end framework for multivariate time series forecastingunder variable missingness. The model introduces an adaptive missing-value filling and enhancement layer with learnable embeddings to mitigate the bias introduced by fixed-value imputation. Inaddition, a dynamic graph convolutional gated recurrent unit is employed to capture temporal and spatial dependencies.

However, two reviewers expressed concerns regarding the limited novelty of the proposed method, noting that the overall framework largely consists of existing modules. The design and combination of these components lack sufficient theoretical justification, and the paper does not clearly explain how each module specifically contributes to addressing the variable-missingness problem targeted by the authors.

Other reviewers pointed out the ablation studies are insufficient, lacking a thorough analysis of the effects of newly introduced embeddings, random masking strategies, and performance under commonly encountered missing-data patterns.
In addition, the manuscript contains multiple issues related to grammar and formatting, which affect its overall clarity and readability. Several model parameters and symbols are insufficiently defined,making parts of the methodology difficult to follow.
Overall, I find that the concerns raised by the reviewers are valid and remain not fully resolved.

Therefore, my recommended decision for this paper is reject.

**Reviewer Concerns:**

Issues related to grammar errors, formatting inconsistencies, and the insufficiency of the ablation study design were partially addressed in the rebuttal.

However, several key concerns raised by the reviewers remain insufficiently resolved, including the unclear definition of parameters, the limited methodological novelty, and how the proposed modeldesign specifically addresses the variable-missingness problem. For example, in Equation (11), the authors combine different representations through simple concatenation, but do not explain why this fusion strategy is preferred over alternative designs, nor how the information learned by E_p is utilizedin subsequent computations. Although the authors emphasize the role of the embedding design in mitigating error accumulation, the mechanism by which errors are reduced or prevented frompropagating is not clearly articulated. From an overall architectural perspective, aside from the embedding layer, the remaining network components are highly similar to existing STGNN frameworks, and the current rebuttal does not suffi ciently demonstrate the methodological novelty ofthe proposed approach.

**Reviewer Scores:**

As no reviewers responded during the discussion phase and the concerns raised by each reviewer remain not fully resolved, I guess that the reviewers would have maintained their original scores.

---

### Decision · Program_Chairs · 2026-01-26

Reject